# LSD1-mediated enhancer silencing attenuates retinoic acid signalling during pancreatic endocrine cell development

Nicholas K. Vinckier[1,6], Nisha A. Patel[1,6], Ryan J. Geusz [1,6], Allen Wang[1], Jinzhao Wang [1], Ileana Matta[1], Austin R. Harrington[1], Matthew Wortham[1], Nichole Wetton[1], Jianxun Wang[2], Ulupi S. Jhala[3], Michael G. Rosenfeld [2], Christopher W. Benner [4], Hung-Ping Shih [5] & Maike Sander [1✉]

Developmental progression depends on temporally defined changes in gene expression mediated by transient exposure of lineage intermediates to signals in the progenitor niche. To determine whether cell-intrinsic epigenetic mechanisms contribute to signal-induced transcriptional responses, here we manipulate the signalling environment and activity of the histone demethylase LSD1 during differentiation of hESC-gut tube intermediates into pancreatic endocrine cells. We identify a transient requirement for LSD1 in endocrine cell differentiation spanning a short time-window early in pancreas development, a phenotype we reproduced in mice. Examination of enhancer and transcriptome landscapes revealed that LSD1 silences transiently active retinoic acid (RA)-induced enhancers and their target genes. Furthermore, prolonged RA exposure phenocopies LSD1 inhibition, suggesting that LSD1 regulates endocrine cell differentiation by limiting the duration of RA signalling. Our findings identify LSD1-mediated enhancer silencing as a cell-intrinsic epigenetic feedback mechanism by which the duration of the transcriptional response to a developmental signal is limited.

[1] Departments of Pediatrics and Cellular & Molecular Medicine, Pediatric Diabetes Research Center, Sanford Consortium for Regenerative Medicine, University of California, San Diego, La Jolla, CA 92093, USA. [2] Howard Hughes Medical Institute and Department of Medicine, University of California, San Diego, La Jolla, CA 92093, USA. [3] Department of Pediatrics and Pediatric Diabetes Research Center, University of California, San Diego, La Jolla, CA 92093, USA. [4] Department of Cellular & Molecular Medicine, University of California, San Diego, La Jolla, CA 92093, USA. [5] Department of Translational Research & Cellular Therapeutics, Diabetes & Metabolism Research Institute, City of Hope, Duarte, CA 91010, USA. [6]These authors contributed equally: Nicholas K. Vinckier, Nisha A. Patel, Ryan J. Geusz. ✉email: masander@ucsd.edu

During development, intermediate progenitors progress toward a distinct cell fate as a result of sequential instructions by signalling cues in the progenitor niche. The duration of a developmental signal has to be limited in order for developmental intermediates to appropriately respond to the next inductive cue. For example, pancreas induction from the foregut endoderm requires retinoic acid (RA) signalling, but thereafter RA signalling activity needs to be dampened for pancreatic progenitors to correctly interpret pro-endocrine differentiation cues[1]. Thus, signalling cues are interpreted in a highly context-dependent manner and signals need to be temporally limited to delineate critical competence windows for developmental transitions. An open question is whether removal of the signal is sufficient to terminate a response to a signal or whether cell-intrinsic mechanisms at the level of the responder tissue enable developmental transitions by limiting the duration of signal-induced transcriptional responses.

Spatiotemporal gene expression during development is regulated by transcriptional enhancers[2]. Chromatin state at enhancers is a significant determinant of transcriptional responsiveness to environmental signals, and enhancers respond to signalling cues by modifying their chromatin state. Enhancers can exhibit an inactive, poised, or active chromatin state. Inactive enhancers are characterised by compact chromatin and absence of active histone modifications, whereas poised enhancers are nucleosome-free and marked by mono- and di-methylation of histone H3 at lysine 4 (H3K4me1 and H3K4me2)[3–5]. The transition from an inactive to a poised enhancer state during development coincides with a gain in cellular competence of lineage intermediates to respond to inductive signalling cues[5]. Thus, developmental competence can be defined as a temporal state during which the epigenetic landscape is permissive for responding to environmental signals. Signal-dependent transcription factors (TFs) activate poised enhancers by recruiting co-activator complexes containing histone acetyltransferases (HATs) that deposit H3K27 acetylation (H3K27ac) marks, thereby transforming the poised enhancer into one that actively supports transcription[6]. It is unknown whether or not the erasure of these epigenetic marks is a prerequisite for termination of one competence window and transition to the next.

Lysine-specific demethylase 1 (LSD1), also known as KDM1A, regulates chromatin by catalysing the removal of mono- and di-methyl marks from K4 at histone H3[7], thus rendering poised enhancer chromatin inactive[8]. This process has been called enhancer decommissioning and is coupled to complete silencing of associated genes[8]. Despite its role in enhancer silencing, LSD1 frequently resides in complexes of active enhancers[9–11]. In the context of acetylated histones, LSD1 activity and demethylation of H3K4 is inhibited[9]. Therefore, current evidence suggests that histones need to be deacetylated before LSD1 can decommission active enhancers. Consistent with this mechanism, LSD1 occupies enhancers of pluripotency genes in pluripotent stem cells and decommissions these enhancers only when pluripotent stem cells undergo differentiation[8]. Whether LSD1-mediated regulation of enhancer chromatin plays a role in defining developmental competence windows and enabling sequential cell state transitions remains unknown.

Here, we ask whether epigenetic mechanisms can limit the duration of an inductive signal throughout a developmental time course, thereby defining distinct competence windows and preventing inappropriate responses to developmental signals. To investigate this, we manipulate LSD1 activity and RA signalling in a human embryonic stem cell (hESC)-based differentiation system, where cells progress stepwise in defined conditions toward the pancreatic endocrine cell fate. We show that LSD1-mediated enhancer decommissioning limits the time window during which cells express RA-induced genes. When LSD1 activity is inhibited

immediately after pancreas induction, RA-induced genes fail to be silenced despite removal of RA as an inductive signal, which is associated with an inability of the cells to undergo endocrine cell differentiation. Thus, our results show that loss of LSD1 function critically alters the epigenetic landscape that terminates the competence window for RA signalling. These findings identify modification of the epigenome as an important cell-intrinsic mechanism for sharpening transcriptional responses to developmental signals.

## Results

### Human pancreatic endocrine cell development requires LSD1.

To investigate possible roles for LSD1 during defined windows of transition to a differentiated cell type, we employed a hESC differentiation system, in which cells progress stepwise toward the pancreatic endocrine cell lineage through sequential exposure to signalling cues that guide corresponding cell state transitions in the developing embryo (Fig. 1a)[5,12–14]. In this differentiation system, LSD1 was broadly expressed throughout progression to the endocrine cell stage (EN) (Supplementary Fig. 1a, b). Likewise, levels of flavin adenine dinucleotide (FAD), which is a metabolic cofactor of LSD1[15], did not change substantially throughout the differentiation time course (Supplementary Fig. 1c). We verified LSD1 expression in pancreatic progenitor cells and differentiated endocrine cells in human foetal and adult tissue (Supplementary Fig. 1d).

To assess whether LSD1 is required for pancreatic development, we started by blocking LSD1 activity immediately after the initiation of pancreas induction during the transition from the early (PP1) to the late (PP2) pancreatic progenitor cell stage (LSD1i*early*), using the irreversible LSD1 inhibitor tranylcypromine (TCP) (Fig. 1a). PP1 and PP2 progenitors are distinguished by increasing expression of pancreatic TFs that commit progenitors to the endocrine cell fate, including NKX6.1 and NGN3[12]. Thus, PP1 cells represent a less committed pancreatic progenitor cell stage, whereas PP2 cells exhibit features of endocrine cell commitment. LSD1 inhibition during the PP1 to PP2 transition did not negatively affect expression of PDX1, NKX6.1, or NGN3 (Supplementary Fig. 1e–h), indicating that endocrine-committed pancreatic progenitors can form in the absence of LSD1. However, when LSD1i*early* cells were further differentiated to the EN stage, we observed a striking absence of endocrine cells at the EN stage, while progenitor cell markers remained largely unaffected (Fig. 1b–d and Supplementary Fig. 2). The same phenotype was observed when culturing in the presence of several other irreversible and reversible LSD1 inhibitors during the PP1 to PP2 transition or by transducing cells with a lentivirus expressing shRNAs for *LSD1* a day prior to the PP1 stage (Supplementary Figs. 3a–d and 4a–c). The normal progression through endocrine commitment but the absence of endocrine cells after LSD1 inhibition indicated a specific requirement for LSD1 activity during endocrine cell differentiation. To directly test whether the endocrine cell differentiation step requires LSD1 activity, we added TCP or the LSD1 inhibitor GSK2879552 during the PP2 to EN transition (LSD1i*late*). Surprisingly, this later inhibition of LSD1 did not perturb endocrine cell formation (Fig. 1b–d and Supplementary Fig. 3b–d). Thus, endocrine cell development requires LSD1 activity during a narrow time window after pancreas induction, but not during endocrine cell differentiation. This indicates that LSD1-mediated changes during the PP1 to PP2 transition affect the ability of developmental precursors to undergo endocrine differentiation later in development.

### LSD1 inhibition prevents enhancer silencing.

Given LSD1's role as a chromatin modifier[7], we investigated whether loss of LSD1

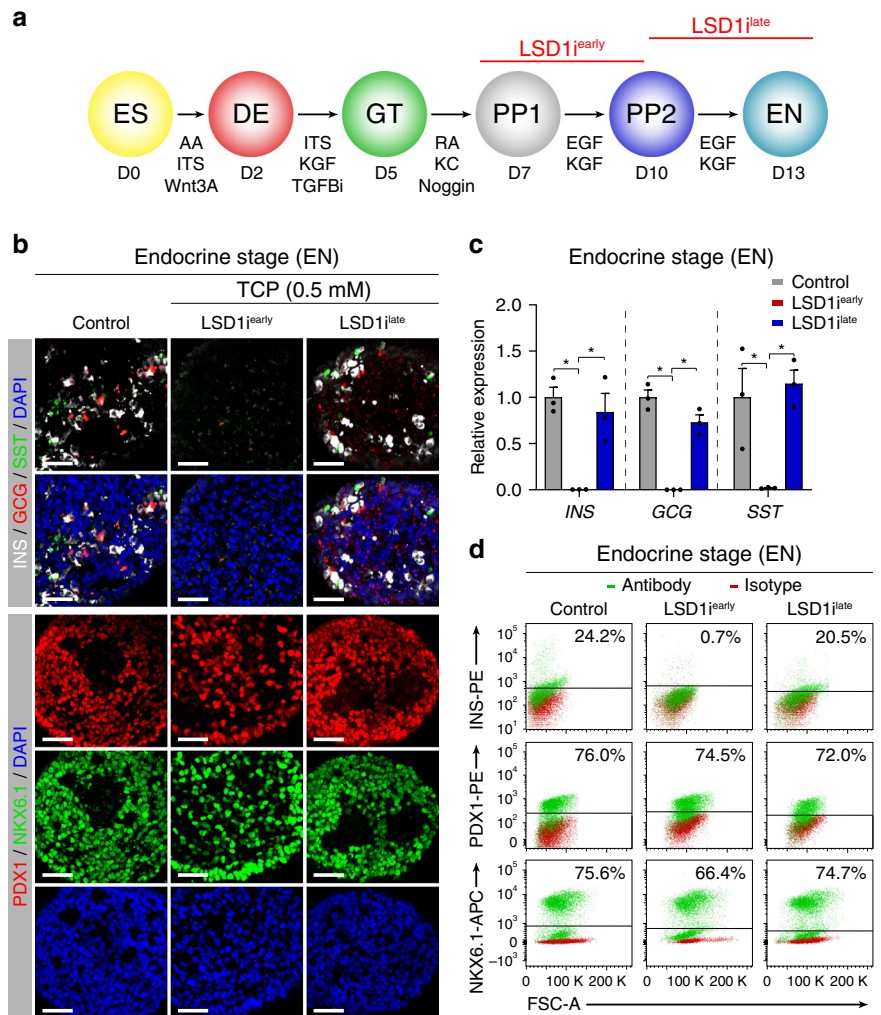

**Fig. 1 Endocrine cell formation requires LSD1 activity during a short window in early pancreatic development. a** Schematic of the human embryonic stem cell (hESC) differentiation protocol to the endocrine cell stage (EN) and experimental plan for LSD1 inhibition. **b** Immunofluorescent staining for pancreatic hormones insulin (INS), glucagon (GCG) and somatostatin (SST) or PDX1 and NKX6.1 in control EN cells compared to EN cells with early (LSD1i*early*) and late (LSD1i*late*) LSD1 inhibition (representative images, $n = 10$ independent differentiations). Scale bar, 50 μm. **c** qRT-PCR analysis for *INS*, *GCG* and *SST* in control, LSD1i*early* and LSD1i*late* EN cells. Data are shown as mean ± S.E.M. ($n = 3$ replicates from independent differentiations with $n = 3$ technical replicates per sample; source data are provided as a Source Data file). $P = 7.93\ e{-}4$, $1.42\ e{-}2$, $2.32\ e{-}4$, $8.71\ e{-}4$, $3.5\ e{-}2$, and $1.52\ e{-}3$, respectively, Student's *t*-test, 2 sided. **d** Flow cytometry analysis at EN stage for NKX6.1, PDX1 and INS comparing control, LSD1i*early* and LSD1i*late* cells. Isotype control for each antibody is shown in red and target protein staining in green. Percentage of cells expressing each protein is indicated (representative experiment, $n = 2$ independent differentiations). D, day; AA, activin A; ITS, insulin-transferrin-selenium; TGFBi, TGFβ R1 kinase inhibitor; KC, KAAD-cyclopamine; KGF, keratinocyte growth factor; RA, retinoic acid; EGF, epidermal growth factor; ES, human embryonic stem cells; DE, definitive endoderm; GT, primitive gut tube; PP1, early pancreatic progenitors; PP2, late pancreatic progenitors; EN, endocrine cell stage; FSC-A, forward scatter area.

activity during the PP1 to PP2 transition could block endocrine cell development due to aberrant regulation of the epigenome. To this end, we performed chromatin immunoprecipitation sequencing (ChIP-seq) for LSD1 at the PP1 stage and mapped chromatin state changes at LSD1-bound sites during the PP1 to PP2 transition without and with LSD1 inhibition. We identified a total of 15,084 LSD1 peaks at the PP1 stage throughout the genome (Supplementary Fig. 5a and Supplementary Data 1, 2). Of these, the vast majority were promoter-distal (11,799; >3 kb from TSS; Supplementary Fig. 5a and Supplementary Data 2), which is consistent with prior observations in hESCs[8]. Distal LSD1 peaks at PP1 overlapped with binding sites for the early pancreatic TFs FOXA1, FOXA2, GATA4, GATA6, and HNF6, suggesting that these TFs reside in a complex with LSD1 (Supplementary Fig. 5b,c).

Distal enhancers are highly dynamic during pancreatic development[5], leading us to postulate that LSD1 controls

endocrine cell differentiation by regulating changes in enhancer chromatin state during the PP1 to PP2 transition. To test this, we performed ChIP-seq for the active enhancer mark H3K27ac[4,16,17] at the PP1 and PP2 stage. Reasoning that effects of LSD1 on the active enhancer landscape would be most likely to affect gene expression and therefore have high propensity to be causal for the phenotype, we isolated enhancers that are active at PP1 and/or PP2 and also bound by LSD1 at the PP1 stage. This analysis revealed three groups of LSD1-bound enhancers: Group 1 (G1) enhancers ($n = 1345$) underwent deactivation during the PP1 to PP2 transition (≥2-fold decrease in H3K27ac); Group 2 (G2) enhancers ($n = 765$) were active at both PP1 and PP2 (<2-fold change in H3K27ac); and Group 3 (G3) ($n = 511$) enhancers underwent activation (≥2-fold increase in H3K27ac) during the PP1 to PP2 transition (Fig. 2a and Supplementary Data 3–5). We next examined the "poised" chromatin modifications H3K4me1

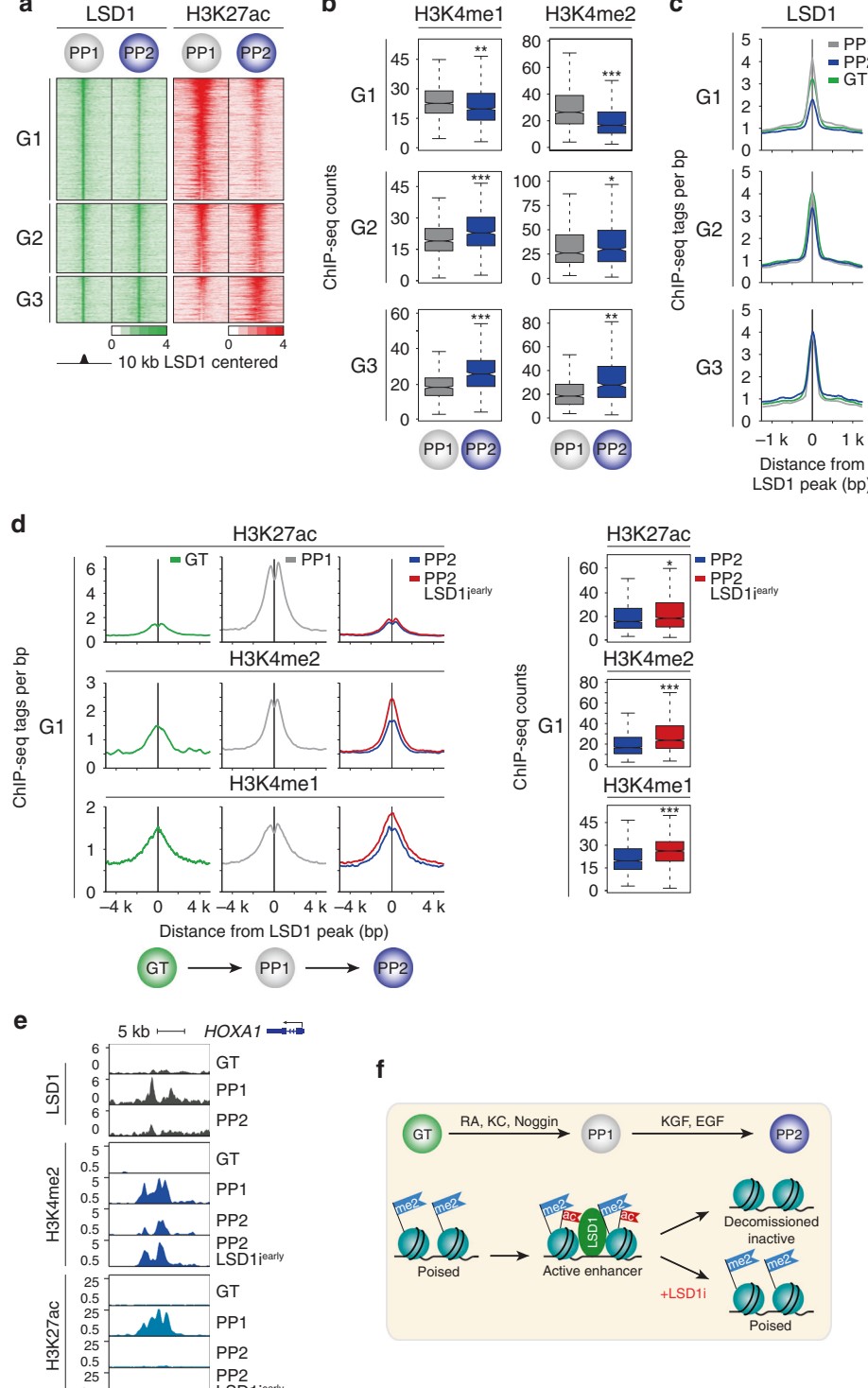

and H3K4me2 at these three enhancer groups during the PP1 to PP2 transition. We observed that LSD1-bound G1 enhancers exhibited a marked decrease in H3K4me1 and H3K4me2 (Fig. 2b), consistent with known roles of LSD1 as a H3K4me2 and H3K4me1 demethylase[7,18]. Thus, G1 enhancers are decommissioned during the PP1 to PP2 transition. To investigate whether LSD1 recruitment is regulated at these decommissioned enhancers, we examined LSD1 occupancy also in gut tube (GT) and PP2 cells. At G1 enhancers, we observed an increase in LSD1 ChIP-seq signal from GT to PP1 and a decrease from PP1 to PP2,

whereas LSD1 signal was similar at all stages in the G2 and G3 enhancer groups (Fig. 2c). Since endocrine cell development requires LSD1 activity during the PP1 to PP2, but not the PP2 to EN transition, the transient recruitment of LSD1 to G1 enhancers at the PP1 stage could signify a specific importance of this enhancer group for the endocrine differentiation phenotype.

To determine whether LSD1 activity is required for remodelling enhancer chromatin during the PP1 to PP2 transition, we analysed H3K27ac, H3K4me2 and H3K4me1 modifications in PP2 cells after LSD1 inhibition (LSD1i$^{early}$). In all three enhancer

**Fig. 2 LSD1 inhibition prevents decommissioning of transiently active early pancreatic enhancers. a** Heatmap showing density of ChIP-seq reads for LSD1 and H3K27ac centred on LSD1 peaks, spanning 10 kb. G1, G2 and G3 groups of LSD1-bound enhancers are deactivated (G1; $n = 1345$), remain active (G2; $n = 765$), or are activated (G3; $n = 511$) from PP1 to PP2. **b** Box plots of H3K4me1 and H3K4me2 ChIP-seq counts at G1, G2 and G3 enhancers at PP1 and PP2 stages. Plots are centred on median, with box encompassing 25th–75th percentile and whiskers extending up to 1.5 interquartile range (Tukey style). $P = 5.0$ e$-12$, $< 2.2$ e$-16$, $<2.2$ e$-16$, $1.73$ e$-2$, $<2.2$ e$-16$, and $8.23$ e$-14$, respectively, Wilcoxon rank-sum test, 2 sided. **c** Tag density plots displaying LSD1 tag distribution at G1, G2 and G3 enhancers at GT, PP1 and PP2 stages, centred on PP1 LSD1 peaks. **d** Tag density plots (left) for G1 enhancers displaying H3K27ac, H3K4me2 and H3K4me1 tag distribution at GT and PP1 stages, and at PP2 stage with and without early LSD1 inhibition (TCP, LSD1i[early]). Plots are centred on PP1 LSD1 peaks. Box plots (right) of H3K27ac, H3K4me2 and H3K4me1 ChIP-seq counts at G1 enhancers at PP2 stage with and without early LSD1 inhibition (LSD1i[early]). Plots are centred on median, with box encompassing 25th–75th percentile and whiskers extending up to 1.5 interquartile range (Tukey style). $P = 4.59$ e$-5$, $<2.2$ e$-16$, and $<2.2$ e$-16$, respectively, Wilcoxon rank-sum test, 2 sided. **e** LSD1, H3K4me2, and H3K27ac ChIP-seq profiles at an **e**nhancer near *HOXA1*. **f** Model for LSD1-dependent enhancer decommissioning. Enhancer deactivation by removal of acetylation from H3K27 occurs independent of LSD1 activity. LSD1 subsequently mediates enhancer decommissioning by removal of H3K4me2 marks. KC, KAAD-cyclopamine; KGF, keratinocyte growth factor; RA, retinoic acid; EGF, epidermal growth factor. GT, primitive gut tube; PP1, early pancreatic progenitors; PP2, late pancreatic progenitors. All ChIP-seq experiments, $n = 2$ replicates from independent differentiations.

clusters, we observed little effect of LSD1 inhibition on H3K27ac dynamics during the PP1 to PP2 transition (Fig. 2d, e and Supplementary Fig. 5d). The activation (i.e. acetylation) of G1 enhancers coincided with the pancreas induction step from GT to PP1 (Fig. 2d, e and Supplementary Fig. 5e). Confirming our prior observation that pancreas-specific enhancers are poised prior to activation[5], G1 enhancers exhibited significant deposition of H3K4me1 at the GT stage (Fig. 2d). Thus, G1 enhancers become activated during pancreas induction and are quickly fully inactivated (i.e. decommissioned) as pancreatic endocrine development proceeds. Consistent with LSD1's enzymatic activity, LSD1 inhibition during the PP1 to PP2 transition led to significant accumulation of H3K4me1 and H3K4me2, particularly at G1 enhancers (Fig. 2d, e and Supplementary Fig. 5e; $p < 2.2$e$-16$, Wilcoxon rank-sum test). H3K4me1 and H3K4me2 levels at G1 enhancers in LSD1i[early] PP2 cells were similar to levels at PP1, showing a requirement for LSD1 in decommissioning these enhancers during the PP1 to PP2 transition. Although H3K4me1 and H3K4me2 levels were also increased at G2 and G3 enhancers after LSD1 inhibition, the effect was less pronounced compared to G1 enhancers (Supplementary Fig. 5d). Importantly, H3K4me1 and H3K4me2 deposition was not increased at enhancers not bound by LSD1 (Supplementary Fig. 5f and Supplementary Data 6), demonstrating specificity of the effect to LSD1-bound enhancers. Combined, this analysis identified a LSD1-regulated set of enhancers that is activated upon addition of pancreas-inductive factors during the GT to PP1 transition and deacetylated and decommissioned (i.e. demethylated) when these factors are withdrawn from PP1 to PP2 (Fig. 2f). We find that deacetylation of these enhancers occurs largely independent of LSD1, but that LSD1 is required for enhancer decommissioning and thus complete enhancer silencing. Given prior findings that LSD1 activity is inhibited in context of acetylated histones[9], these results suggest that histone acetylation from GT to PP1 prevents LSD1-mediated enhancer silencing and that LSD1-independent H3K27ac removal allows LSD1 to silence these enhancers during the PP1 to PP2 transition.

**LSD1 represses retinoic acid-dependent genes**. We next sought to investigate possible effects of the observed chromatin changes on gene expression and compared RNA-seq profiles of control PP2 cells and PP2 cells after LSD1 inhibition (LSD1i[early]). This analysis identified 445 genes that decreased and 955 genes that increased in expression due to LSD1 inhibition (Fig. 3a and Supplementary Data 7 and 8; $p < 0.05$, $\geq 1.5$-fold change). To identify those genes most likely directly regulated by LSD1, we performed enrichment analysis for G1, G2, and G3 enhancers as well as other distal LSD1 binding sites near genes up- and downregulated after LSD1 inhibition (TSS ± 100 kb from LSD1

peak). G1, G2, and G3 enhancers, but not other distal LSD1 binding sites, showed significant enrichment close to genes upregulated due to LSD1 inhibition (Fig. 3b and Supplementary Data 3–6). The majority of the enhancer-associated upregulated genes were near G1 enhancers (Fig. 3c). By contrast, we observed significant depletion or lack of enrichment of distal LSD1-bound sites near genes downregulated in LSD1i[early] cells (Supplementary Fig. 6a). Together, this analysis suggests that direct LSD1 target genes are overrepresented among genes upregulated after LSD1 inhibition, whereas downregulated genes are not directly LSD1-regulated.

We next determined how candidate G1, G2 and G3 enhancer target genes (Supplementary Data 9–11) are regulated over the developmental time course (Fig. 3d and Supplementary Fig. 6b, c). G1 enhancer-associated genes that were upregulated by LSD1 inhibition were induced during the GT to PP1 transition and then downregulated during the transition to PP2 (Fig. 3d). Thus, the expression pattern of G1 enhancer-associated, LSD1-regulated genes mirrors the acetylation pattern of G1 enhancers, which are not acetylated at the GT stage, acetylated at the PP1 stage, and LSD1-dependently decommissioned during the PP1 to PP2 transition (Fig. 2d). Unbiased analysis of over-represented pathways among genes upregulated by LSD1 inhibition revealed enrichment for genes linked to RA signalling in the G1, but not G2 or G3, enhancer-associated group of genes, suggesting an important role for RA signalling in the regulation of G1 enhancer-associated genes (Fig. 3e, Supplementary Fig. 6b, c and Supplementary Data 12–14). To simulate the requirement for RA signalling in pancreatic lineage induction in vivo[19,20], RA is one of three growth factors added to the culture medium during the GT to PP1 transition to induce pancreatic genes (Fig. 1a)[12]. RA is subsequently withdrawn during the transition from PP1 to PP2. During the PP1 to PP2 transition, the only possible source for stimulation of the RA receptor (RAR) are traces of retinol in the B27 supplement. Thus, the activity of G1 enhancers and expression of associated genes precisely coincides with the addition and removal of exogenous RA.

**Prolonged retinoic acid exposure phenocopies LSD1 inhibition**. RA regulates gene expression by binding to its hetero-dimeric receptor composed of RAR and retinoid X receptor (RXR)[21]. In the absence of RA, the RAR/RXR heterodimer recruits co-repressors leading to histone deacetylation and gene silencing, while RA binding to RAR/RXR induces recruitment of HATs, mediating histone acetylation and activation of RA-dependent genes. Hence, the observed pattern of H3K27 acetylation at G1 enhancers during progression from GT to PP2 (Fig. 2d) is consistent with RA-dependent regulation of these enhancers. To determine whether G1 enhancers are indeed

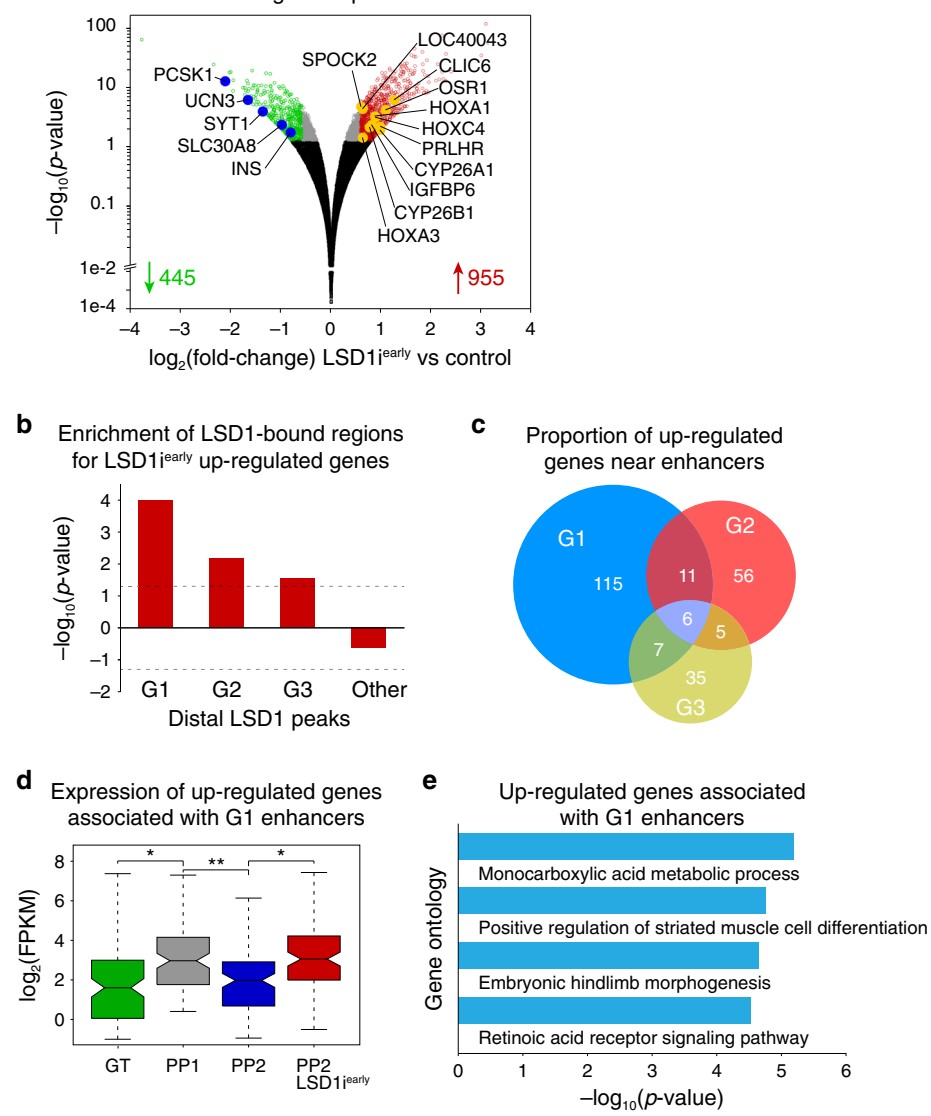

**Fig. 3 LSD1 activity is necessary for repressing transiently expressed retinoic acid-dependent genes. a** Volcano plot of differentially expressed genes at PP2 after LSD1 inhibition from PP1 to PP2 (TCP, LSD1i$^{early}$). Differential expression calculated with DESeq2 and genes with ≥1.5-fold change up or down. Adjusted *p*-values of <0.05 were considered differentially expressed. 445 genes were downregulated and 955 were upregulated in LSD1i$^{early}$ PP2 cells. Black dots indicate genes not significantly changed (*p*-value > 0.05), grey dots genes significantly changed (*p*-value < 0.05) but less than 1.5-fold compared to control, red and green dots genes significantly up- and downregulated (*p*-value < 0.05 and ≥1.5-fold change), respectively (*n* = 2 replicates from independent differentiations). **b** Enrichment analysis of genes upregulated by LSD1i$^{early}$ within 100 kb of G1, G2 and G3 or other distal LSD1 peaks. Dashed lines indicate *p*-value = 0.05 for enrichment (positive value) or depletion (negative value), permutation test. **c** Percentage of LSD1i$^{early}$ upregulated genes near G1 (*n* = 139), G2 (*n* = 78) and G3 (*n* = 53) enhancers (within 100 kb). **d** Box plot of mRNA levels for 139 LSD1i$^{early}$ upregulated genes near G1 enhancers. Plots are centred on median, with box encompassing 25th–75th percentile and whiskers extending up to 1.5 interquartile range (Tukey style). *P* = 2.30 e−3, 4.38 e−6, and 2.25 e−7, respectively, Wilcoxon rank-sum test, 2 sided. **e** Gene ontology analysis for 139 LSD1i$^{early}$ upregulated genes near G1 enhancers. GT, primitive gut tube; PP1, early pancreatic progenitors; PP2, late pancreatic progenitors.

regulated by RA, we performed TF binding motif enrichment analysis. This analysis revealed significant enrichment of the motif for the RAR/RXR heterodimer at G1 compared to G2 and G3 enhancers (Fig. 4a and Supplementary Data 15). When motifs at G2 and G3 enhancers were compared against the entire genome, excluding G1, G2, and G3 regions, no RAR/RXR motif enrichment was observed, further supporting specific enrichment of the RAR/RXR motif at G1 enhancers (Supplementary Data 16). ChIP-seq analysis for RXR, which is the obligatory binding partner for all RAR isoforms, confirmed RXR binding to G1 enhancers at the PP1 stage (Fig. 4b, Supplementary Fig. 5c and Supplementary Data 17). RXR binding was enriched at G1

enhancers when compared to either G2 or G3 enhancers (Fisher's exact test, *p* = 1.728 e−8 and p = 9.427 e−15, respectively), indicating RA-dependent regulation particularly of G1 enhancers.

To determine whether failure to silence RA-induced genes could be the mechanism by which LSD1 inhibition blocks endocrine cell differentiation, we tested whether extended RA exposure of pancreatic progenitors abrogates endocrine cell differentiation in a similar manner as LSD1 inhibition. To extend the time period of RA signalling, we added the RA analogue TTNPB not only from the GT to PP1 stage, but also during the PP1 to PP2 transition (Fig. 4c, RA$^{extended}$) and then differentiated RA$^{extended}$ cultures to the EN stage. Mimicking the LSD1i$^{early}$

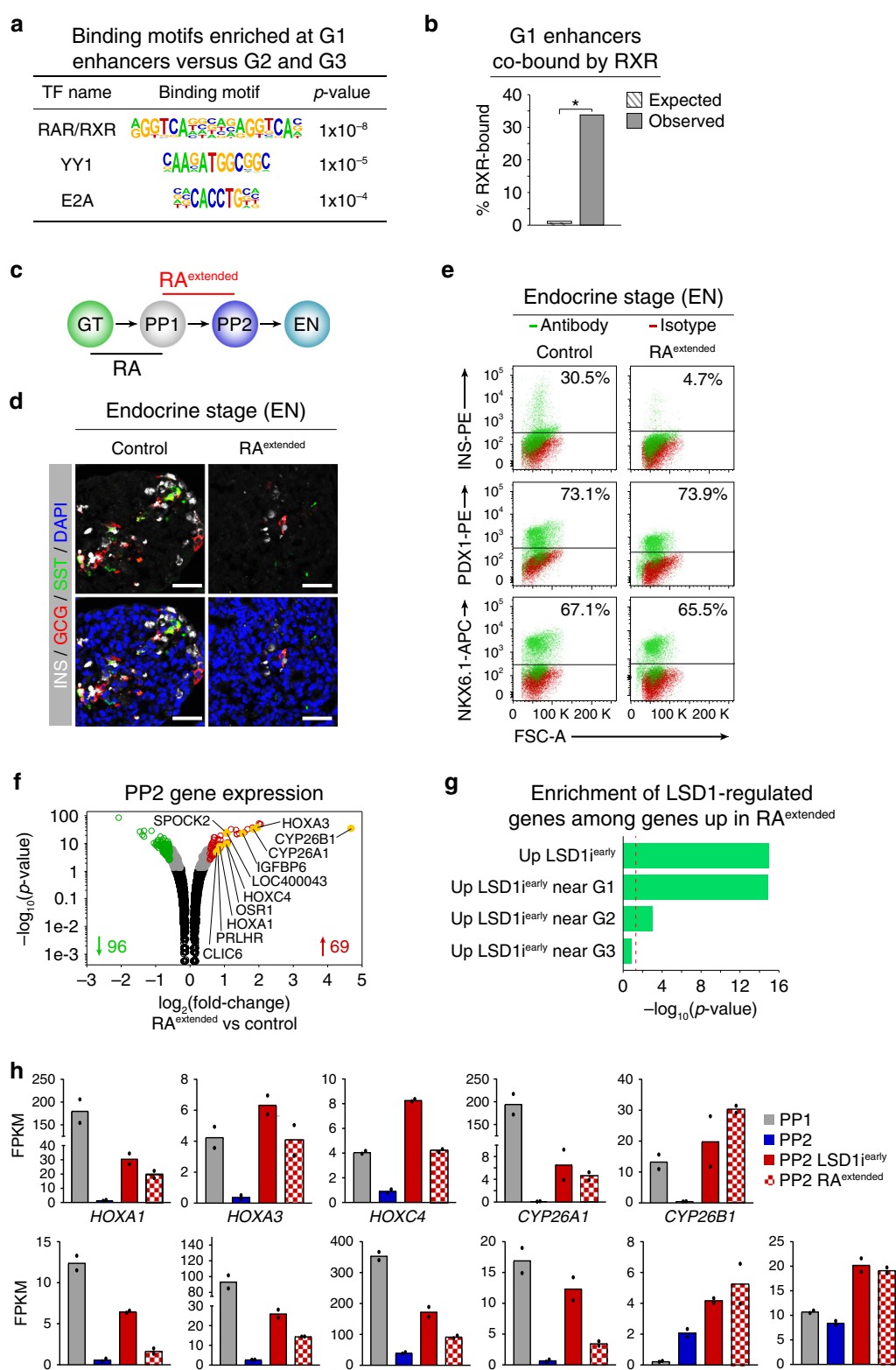

**a** Binding motifs enriched at G1 enhancers versus G2 and G3

| TF name | Binding motif | *p*-value |
|---|---|---|
| RAR/RXR | | $1\times10^{-8}$ |
| YY1 | | $1\times10^{-5}$ |
| E2A | | $1\times10^{-4}$ |

**b** G1 enhancers co-bound by RXR

**c** RA$^{extended}$  GT → PP1 → PP2 → EN  RA

**d** Endocrine stage (EN)  Control   RA$^{extended}$  INS / GCG / SST / DAPI

**e** Endocrine stage (EN)  — Antibody  — Isotype  Control   RA$^{extended}$

**f** PP2 gene expression

**g** Enrichment of LSD1-regulated genes among genes up in RA$^{extended}$

**h**

phenotype, EN stage RA$^{extended}$ cultures exhibited a striking absence of endocrine cells, while the progenitor cell markers PDX1, NKX6.1, and NGN3 were unaffected (Fig. 4d, e and Supplementary Fig. 7a–d). Thus, RA exposure of pancreatic progenitors has to be transient for endocrine cell differentiation to occur.

We next employed RNA-seq analysis to identify genes dysregulated as a result of prolonged RA exposure. Only 96 genes were downregulated and 69 upregulated in RA$^{extended}$ PP2 cultures (Fig. 4f and Supplementary Data 18 and 19; $p < 0.05$, ≥1.5-fold change), suggesting that the endocrine differentiation block is mediated by dysregulation of a modest number of genes. Consistent with the small number of dysregulated genes in RA$^{extended}$ cells compared to LSD1i$^{early}$ cells, RA$^{extended}$ cells more closely resembled untreated PP2 cells than LSD1i$^{early}$ cells (Supplementary Fig. 6e). Strikingly, genes upregulated after LSD1

**Fig. 4 Prolonged retinoic acid exposure of early pancreatic progenitor cells phenocopies LSD1 inhibition. a** Enriched transcription factor (TF) binding motifs with associated p-values for G1 enhancers compared to G2 and G3 enhancers. Fisher's exact test, 2 sided, corrected for multiple comparisons. **b** Enrichment for RXR peaks (±1 kb) among G1 enhancers versus random genomic regions. $P = 0$, permutation test. **c** Experimental plan to extend retinoic acid (RA) exposure through PP1 to PP2 (RA$^{extended}$) during hESC differentiation to the endocrine cell stage (EN). **d** Immunofluorescent staining for insulin (INS), glucagon (GCG) and somatostatin (SST) in control EN cells compared to EN cells with extended RA treatment (RA$^{extended}$) (representative images, $n = 3$ independent differentiations). Scale bar, 50 μm. **e** Flow cytometry analysis at EN stage for NKX6.1, PDX1 and INS comparing control and RA$^{extended}$ cultures. Isotype control for each antibody is shown in red and target protein staining in green. Percentage of cells expressing each protein is indicated (representative experiment, $n = 2$ independent differentiations). **f** Volcano plot of differentially expressed genes at PP2 in RA$^{extended}$ cultures. Differential expression calculated with DESeq2 and genes with ≥1.5-fold change up or down. Adjusted p-values < 0.05 were considered differentially expressed. 96 genes were downregulated and 69 were upregulated in RA$^{extended}$ cultures. Black dots indicate genes not significantly changed (p-value > 0.05), grey dots genes significantly changed (p-value < 0.05) but less than 1.5-fold compared to control, red and green dots genes significantly up- and downregulated (p-value < 0.05 and ≥ 1.5-fold change), respectively. Yellow dots highlight genes also upregulated after LSD1 inhibition from PP1 to PP2 (TCP, LSD1i$^{early}$) ($n = 2$ replicates from independent differentiations). **g** Enrichment analysis of genes associated with LSD1-bound enhancers and upregulated by LSD1i$^{early}$ among those upregulated by RA$^{extended}$. Dashed line indicates p-value = 0.05, Fisher's exact test, 2 sided. **h** mRNA levels of select genes significantly upregulated in both LSD1i$^{early}$ and RA$^{extended}$ PP2 cells. Levels at PP1 stage are also displayed. Data shown as mean FPKM ± S.E.M. ($n = 2$ replicates from independent differentiations; source data are provided as a Source Data file). GT, primitive gut tube; PP1, early pancreatic progenitors; PP2, late pancreatic progenitors.

inhibition were significantly enriched among the genes also increased in expression after prolonged RA exposure (Fig. 4f), and genes near G1 enhancers largely accounted for this enrichment (Fig. 4g). Genes downregulated after LSD1 inhibition were likewise enriched among genes decreased in expression after prolonged RA, but LSD1-occupied enhancers were not enriched in the vicinity of these genes (Supplementary Fig. 7f), suggesting indirect effects. Among the genes upregulated by both LSD1 inhibition and prolonged RA exposure were numerous genes known to be regulated by RA, including genes encoding HOX TFs and RA-inactivating enzymes of the CYP26 family (Fig. 4h), supporting the notion that RA-induced genes need to be silenced for cells to acquire competence for endocrine cell differentiation. Together, our findings support a model whereby LSD1 silences RA-regulated genes by decommissioning their enhancers upon RA withdrawal, thereby ensuring transient, ligand-dependent expression of RA-induced genes. Under conditions of prolonged RA exposure, RA-mediated recruitment of HATs maintains histone acetylation[21], which inhibits LSD1 activity[9] and prevents enhancer decommissioning during the PP1 to PP2 transition.

**LSD1 dampens future responses to retinoic acid.** We sought to further substantiate that LSD1 regulates RA responsiveness and that the block in endocrine cell differentiation after LSD1 inhibition is linked to aberrant expression of RA-dependent genes. We predicted that endocrine cell differentiation should not be perturbed when cells are re-exposed to RA during the PP2 to EN transition, because enhancers of early RA-responsive genes are already decommissioned at the PP2 stage (Fig. 2d, e and Supplementary Fig. 5d). To test this, we re-introduced RA into the culture medium during the PP2 to EN transition (RA$^{late}$; Fig. 5a). As hypothesised, and in stark contrast to RA$^{extended}$ cultures (Fig. 4d, e), endocrine cells were present in RA$^{late}$ EN stage cultures in numbers almost identical to control cultures (Fig. 5b, c and Supplementary Fig 8a, b). Thus, similar to addition of the LSD1 inhibitor (Fig. 1a–d), addition of RA prevents endocrine cell formation only during the PP1 to PP2 but not the PP2 to EN transition.

To further test whether enhancer decommissioning is a mechanism by which to regulate RA responsiveness, we re-exposed cells to RA from PP2 to EN (RA$^{late}$) with or without prior LSD1 inhibition during the PP1 to PP2 transition (Fig. 5a). As expected, LSD1i$^{early}$ + RA$^{late}$ treatment completely blocked endocrine cell differentiation, phenocopying LSD1i$^{early}$ cultures (Supplementary Fig. 8c, d). To determine whether prior LSD1 inhibition alters the extent to which RA-regulated genes can be

induced by RA, we measured gene expression changes in response to RA. To this end, we compared gene expression at the EN stage in LSD1i$^{early}$ vs. LSD1i$^{early}$ + RA$^{late}$ and control vs. RA$^{late}$ conditions. We used LSD1i$^{early}$ EN cells, rather than control EN stage cultures, as a reference for the cells re-exposed to RA after LSD1i$^{early}$ treatment to control for the population bias caused by the lack of endocrine cells after LSD1 inhibition. Most genes near G1 enhancers that are upregulated by both LSD1 inhibition (Fig. 3a) and extended RA exposure (Fig. 4f), including HOXA1, HOXA3, HOXC4, and CYP26B1, exhibited a higher degree of inducibility by RA with prior LSD1 inhibition (Fig. 5d, Supplementary Fig. 8e and Supplementary Data 20 and 21). Thus, LSD1 appears to dampen, although not obliterate, future RA responsiveness in cells that have been previously exposed to RA. We observed no difference in the expression of LSD1, RARs, or RXRs between the different conditions (Supplementary Fig. 8f), indicating that alteration of the epigenetic state rather than differences in TF and co-factor expression explain the heightened RA responsiveness after LSD1 inhibition. We note that other factors must also control RA responsiveness since HOXA1, HOXA3, HOXC4, CYP26A1, and CYP26B1 are still induced by late RA treatment without prior LSD1 inhibition.

**Lsd1 is required for endocrine cell development in vivo.** To verify our in vitro findings in an in vivo model, we deleted Lsd1 conditionally in mice to determine whether endocrine cell differentiation requires Lsd1 activity transiently in early pancreas development, as observed in the hESC differentiation system. As in the human pancreas (Supplementary Fig. 1d), Lsd1 was highly expressed in pancreatic progenitors and endocrine cells (Supplementary Fig. 9a, b). To selectively inactivate Lsd1 in early pancreatic progenitors similar to LSD1 inhibition at PP1, we generated Pdx1Cre;Lsd1$^{flox/flox}$ (Lsd1$^{Δpan}$) mice (Fig. 6a). In Lsd1$^{Δpan}$ embryos, key aspects of early pancreatic development, such as the induction of early pancreatic markers and outgrowth of the tissue buds, were unperturbed (Fig. 6b, c and Supplementary Fig. 9c). However, by embryonic day (e) 15.5, when widespread endocrine cell differentiation was evident in control mice, Lsd1$^{Δpan}$ embryos exhibited an almost complete lack of endocrine cells (Fig. 6b), a phenotype that remained apparent at postnatal day (P) 0 (Fig. 6b, d). In vivo inactivation of Lsd1 further revealed that Lsd1 activity is selectively required for the development of the endocrine lineage, while being dispensable for exocrine cell formation and key aspects of early pancreatic development, such as maintenance of pancreatic progenitors and growth of the developing organ (Fig. 6b, c and Supplementary

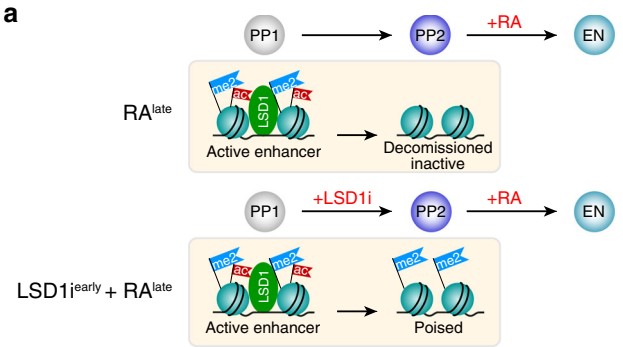

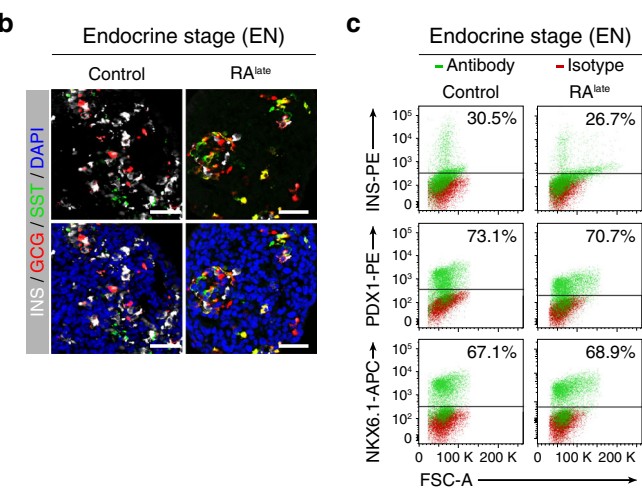

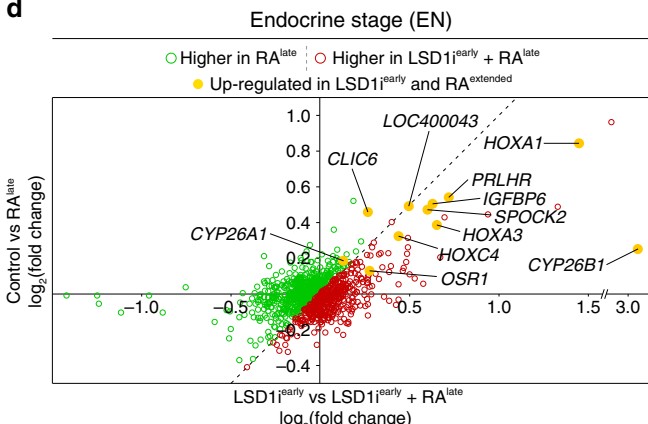

**Fig. 5 LSD1 decreases future inducibility of RA-dependent genes by retinoic acid. a** Experimental plan to re-introduce retinoic acid (RA) during the PP2 to endocrine (EN) transition of hESC differentiation without (RA$^{late}$) and with prior inhibition of LSD1 (LSD1i$^{early}$ + RA$^{late}$). The cartoon depicts the chromatin state as determined in Fig. 2. **b** Immunofluorescent staining for insulin (INS), glucagon (GCG) and somatostatin (SST) in control EN cells compared to EN cells with late RA treatment (RA$^{late}$) (representative images, $n = 3$ independent differentiations). Scale bar, 50 μm. **c** Flow cytometry analysis at EN stage for NKX6.1, PDX1 and INS comparing control and RA$^{late}$ cells. Isotype control for each antibody is shown in red and target protein staining in green. Percentage of cells expressing each protein is indicated (representative experiment, $n = 2$ independent differentiations). **d** Gene expression changes in RA$^{late}$ vs control EN cells compared to gene expression changes in LSD1i$^{early}$ + RA$^{late}$ vs LSD1i$^{early}$ EN cells (calculated with DESeq2). Green dots indicate genes more increased in RA$^{late}$ and red dots indicate genes more increased in LSD1i$^{early}$ + RA$^{late}$ compared to respective controls. Yellow dots highlight genes upregulated after both LSD1i$^{early}$ and extended RA treatment (RA$^{extended}$) from PP1 to PP2. ($n = 2$ replicates from independent differentiations per condition). PP1, early pancreatic progenitors; PP2, late pancreatic progenitors; FSC-A, forward scatter area.

pancreatic progenitors shortly before endocrine cell differentiation similar to PP2 cells, did not affect endocrine cell formation, as evidenced by the presence of Lsd1-deficient hormone$^+$ cell clusters (Fig. 6f and Supplementary Fig. 9g). Therefore, as in the hESC differentiation system, endocrine cell differentiation in mice requires Lsd1 activity during a narrow time window in early pancreas development. Furthermore, early pancreatic inactivation of *Lsd1* resulted in up-regulation of *HoxA1* transcripts (Fig. 6g) as observed in hESC-PP2 cells (Fig. 3a). Combined our in vitro and in vivo findings support a model whereby LSD1 controls progression to the endocrine cell stage by limiting the duration of early RA signalling through chromatin modification at RA-responsive enhancers.

## Discussion

Proper formation of terminally differentiated cell types requires precise timing, amplitude, and, as suggested by this study, duration of developmental signals. Our findings show that decommissioning (i.e. complete inactivation) of RA-dependent early pancreatic enhancers temporally limits the expression of RA-induced genes after RA is removed, thereby creating a sharp and transient gene expression response to RA. This mechanism can explain how the duration of a transcriptional response to a transient inductive signal is limited during a developmental time course. Furthermore, we observed that enhancer decommissioning dampens future gene inducibility by the same signal. Our findings suggest that developmental competence windows are terminated through erasure of epigenetic marks, providing a mechanistic understanding of why developmental signals evoke context-dependent cellular responses. Underscoring the importance of enhancer silencing for developmental progression, we find that the inability to decommission RA-dependent enhancers and down-regulate RA-induced genes coincides with subsequent failure to initiate endocrine cell differentiation. We propose that LSD1-mediated enhancer decommissioning is a responder tissue-intrinsic mechanism by which perduring transcriptional effects of transient developmental signals are prevented. By helping close competence windows during rapid developmental transitions, this mechanism could help create a conducive state for correct interpretation of the next inductive signal.

Our findings indicate that LSD1-mediated silencing of a subset of RA-dependent early pancreatic genes is necessary for

Fig. 9c–e). Analysis of the endocrine progenitor marker Ngn3 further revealed that endocrine lineage commitment was unaffected in *Lsd1*$^{Δpan}$ embryos (Fig. 6b, c). Thus, as in the hESC differentiation system, *Lsd1* inactivation in early pancreatic progenitors of mice prevents endocrine cell differentiation after endocrine fate commitment.

To determine when precisely Lsd1 is required for endocrine cell development, we crossed *Lsd1*$^{flox/flox}$ and *Pdx1CreER$^{TM}$* mice, allowing for temporally controlled *Lsd1* inactivation in pancreatic progenitors by tamoxifen administration (Fig. 6e). Consistent with the phenotype of *Lsd1*$^{Δpan}$ mice, tamoxifen administration at e10.5 (*Lsd1*$^{Δearly}$ mice) resulted in almost complete absence of endocrine cells (Fig. 6f and Supplementary Fig. 9f). Remaining endocrine cells in *Lsd1*$^{Δearly}$ mice were mostly Lsd1$^+$ due to mosaic deletion (Supplementary Fig. 9f, g). By contrast, tamoxifen injection at e12.5 (*Lsd1*$^{Δlate}$ mice), which targets late

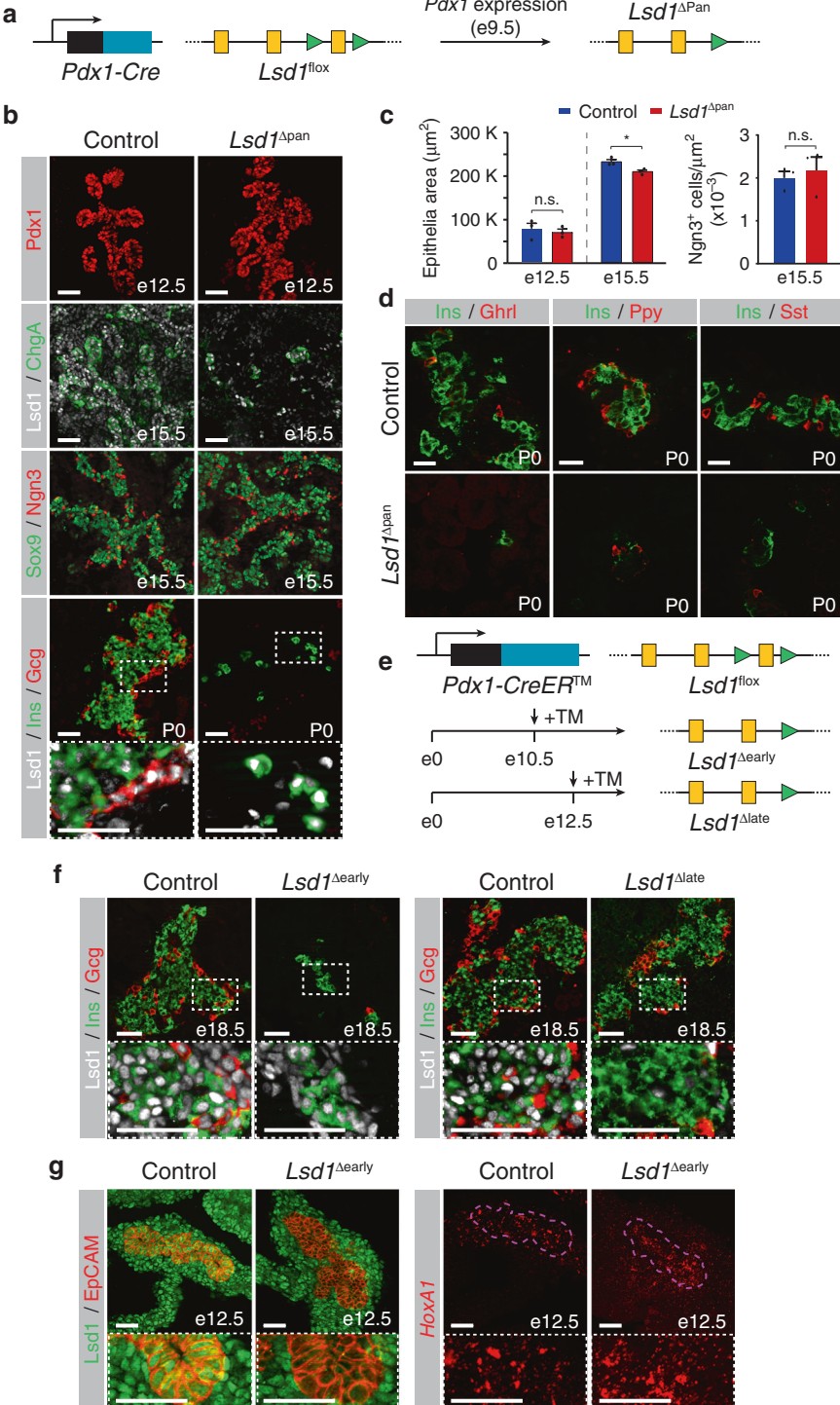

**Fig. 6 Lsd1 is required for endocrine cell formation in mice during a short window in early pancreatic development. a** Strategy for conditional *Lsd1* deletion in embryonic pancreatic progenitors of mice (*Lsd1*^Δpan mice). Yellow boxes: exons; green triangles: *loxP* sites. **b** Immunofluorescent staining for Pdx1 at embryonic day (e) 12.5, Lsd1 and chromogranin A (ChgA) or Sox9 and Ngn3 at e15.5, and Lsd1, insulin (Ins) and glucagon (Gcg) at postnatal day (P) 0 in control and *Lsd1*^Δpan mice (representative images, *n* = 6 embryos per genotype). Boxed areas are shown in higher magnification. Scale bar, 50 μm. **c** Quantification of pancreatic epithelial area at e12.5 and e15.5 and Ngn3+ cells relative to epithelial area. Data are shown as means ± S.E.M. (*n* = 3 embryos per genotype; source data are provided as a Source Data file). *P* = 0.65, 0.02, and 0.39, respectively, Student's t-test, 2 sided. **d** Immunofluorescent staining for Ins with somatostatin (Sst), pancreatic polypeptide (Ppy) and ghrelin (Ghrl) at P0 in control and *Lsd1*^Δpan mice. Scale bar, 25 μm. **e** Strategy for tamoxifen-inducible *Lsd1* deletion in embryonic pancreatic progenitors of mice at e10.5 (*Lsd1*^Δearly) and e12.5 (*Lsd1*^Δlate). Yellow boxes: exons; green triangles: *loxP* sites. **f** Immunofluorescent staining for Lsd1, Ins and Gcg at e18.5 in control, *Lsd1*^Δearly and *Lsd1*^Δlate mice (representative images, *n* = 3 embryos per genotype). Boxed areas are shown in higher magnification. Scale bar, 50 μm. **g** Immunofluorescent staining for Lsd1 and EpCAM (left panels) and RNAscope in situ hybridization for *HoxA1* (right panels, adjacent sections to left panels) in control and *Lsd1*^Δearly embryos at e12.5 (representative images, *n* = 3 *Lsd1*^Δearly embryos and *n* = 6 control embryos). Bottom images show higher magnification. Scale bar, 50 μm.

pancreatic progenitors to acquire competence for endocrine cell differentiation. We identified a set of LSD1-regulated genes, including many *HOX* genes and other genes known to be directly regulated by RA, that are transiently expressed after pancreas induction and quickly downregulated as cells transition to an endocrine-committed pancreatic progenitor cell state, marked by NKX6.1 and NGN3. Several observations suggest that the mechanism by which LSD1 controls endocrine cell development is to silence these RA-regulated genes prior to endocrine cell differentiation. First, we observed that endocrine cell formation in both the hESC-based system and in mice proceeds normally when LSD1 is inhibited or deleted during endocrine cell differentiation, implying that LSD1 has no immediate role in the activation of endocrine genes. Second, like LSD1 inhibition, extended exposure of uncommitted hESC-pancreatic progenitors to RA blocks endocrine cell differentiation, whereas RA exposure during endocrine cell differentiation does not. Similarly, exposure of mouse embryonic pancreatic explants to high RA concentrations prior to endocrine cell differentiation has been shown to impair endocrine cell formation[22]. Furthermore, we showed that after both early LSD1 inhibition and extended RA exposure of hESC-pancreatic progenitors, RA-induced genes are upregulated, which indicates that dysregulation of these genes is relevant for the phenotype. Third, *Lsd1* deletion in pancreatic progenitors of mice has no overt effect on exocrine cell differentiation or other aspects of pancreatic development, suggesting that Lsd1-mediated silencing of these RA-regulated genes occurs specifically during endocrine lineage progression. Consistent with this notion, the expression of HOXA1 is maintained in pancreatic exocrine cells[23]. Similar to pancreas development, LSD1 is expressed in the developing nervous system and plays an important role in neural differentiation[24,25]. Given that RA has time-dependent roles during different phases of neural development[26], it is possible that a similar connection between LSD1 and the regulation of RA signalling windows exists during neurogenesis.

Our findings suggest that the presence of LSD1 in enhancer complexes enables silencing of RA-induced genes after the withdrawal of exogenous RA. In the context of LSD1 inhibition, silencing of RA-dependent genes does not occur despite RA removal, showing that absence of the RA signal is not sufficient for gene inactivation. RA is known to maintain histone acetylation by mediating recruitment of HATs, while removal of RA results in a co-factor switch leading to histone deacetylation[21]. Because presence of acetylated histones inhibits LSD1 activity[9], LSD1-mediated enhancer decommissioning is prevented as long as RA is present, linking enhancer decommissioning and target gene silencing to RA withdrawal. Thus, presence of LSD1 in enhancer complexes provides a cell-intrinsic epigenetic mechanism that couples the duration of gene expression to the presence of the extrinsic signal. It is possible that LSD1 at promoters also contributes to gene regulation after LSD1 inhibition. However, TSSs of the majority of RA-induced genes that were upregulated after LSD1 inhibition (e.g. *HOXA1, HOXA3, HOXC4, CYP26A1, CYP26B1*) were not LSD1-bound (Supplementary Data 1), suggesting that effects on gene expression are mediated by distal enhancers.

We show that inhibition of LSD1 activity during developmental progression coincides with increased H3K4me2/me1 deposition and increased expression of RA-dependent genes after RA removal. This raises the question of whether the catalytic activity of LSD1 and increased H3K4me2/me1 levels or demethylase-independent functions of LSD1 are responsible for the increased expression of RA-dependent genes. Our data suggest that the catalytic activity of LSD1 is indeed important, because TCP and other LSD1 inhibitors inhibit LSD1's catalytic activity. Of note, a recent study in ESCs showed that H3K4me1 is

not required for transcription of nearby genes in the context of acetylated enhancers[27]. Our findings suggest that this is not true in the context of deacetylated enhancers, where H3K4me2/me1 appears to provide an enhancer activation memory that has impact on gene expression. Our observations are consistent with evidence that H3K4me2 deposition is dependent on transcription at enhancers, and that enhancer RNA transcripts correlate with the expression of nearby genes[28]. The exact contexts in which H3K4me2/me1 can drive gene expression and the mechanisms employed will require further studies.

We find that in addition to RAR motifs, LSD1-regulated enhancers are also enriched for FOXA and GATA motifs (Supplementary Data 15) and that FOXA1, FOXA2, GATA4, and GATA6 binding is enriched at LSD1-occupied enhancers (Supplementary Fig. 5b). FOXA1/2 and GATA4/6 have known functions in early pancreas development[29–31]. This indicates that the regulation of early pancreatic enhancers requires collaborative interactions of signal-dependent RARs with other TFs that regulate cell identity. We have previously shown that recruitment of FOXAs to pancreas enhancers precedes pancreas induction, and occurs prior to the addition of RA during hESC differentiation[5]. Before RA is added, FOXA-occupied pancreatic enhancers are poised. Combined with our present findings, this suggests that RA induces the pancreatic lineage by binding to RARs that act upon a set of primed enhancers established by FOXAs and other lineage-determining TFs. FOXAs have previously been shown to broadly prime enhancers of multiple gut tube-derived organs, including liver and lung[5]. While pancreas induction requires RA, lung and liver induction depend on BMP or BMP and WNT signalling, respectively. Therefore, collaboration between signal-dependent TFs and lineage-determining TFs, such as FOXAs, could be a broadly used mechanism to specify different organ lineages from a field of multipotent progenitors. Such mechanism is consistent with the importance of niche signals for specifying progenitor subdomains during development.

One open question is whether the here-described mechanism for LSD1-mediated enhancer silencing is limited to RA-dependent enhancers or could operate also in the context of other signal-dependent enhancers. LSD1 has been shown to reside in transcriptional complexes with signal-dependent TFs of numerous signalling pathways, including the Notch, Wnt, and multiple nuclear hormone receptor signalling pathways[32–34]. Similar to RARs, which associate with HATs upon RA binding[21], the Notch intracellular domain facilitates recruitment of HATs to Notch-responsive enhancers in the presence of ligand, and β-catenin recruits HATs when the Wnt signalling pathway is activated[35,36]. It is therefore conceivable that LSD1 limits the duration of a transcriptional response at these enhancers in a similar manner as shown in this study for RA-responsive enhancers. Such unified mechanism for LSD1 function would explain why LSD1 activity is required in numerous development contexts throughout phylogeny[37–41].

## Methods

**Cell lines.** CyT49 embryonic stem cells were maintained in DMEM F12 (without L-glutamine; VWR) + 10% knockout serum replacement (KSR; Thermo Fisher Scientific), 1% non-essential amino acids (NEAA; Thermo Fisher Scientific), 1% GlutaMAX (Thermo Fisher Scientific), 0.2% β-mercaptoethanol (Thermo Fisher Scientific) and 1% penicillin–streptomycin (Thermo Fisher Scientific). HEK293T were maintained in DMEM F12 containing 100 units/mL penicillin and 100 mg/mL streptomycin sulfate supplemented with 10% foetal bovine serum (FBS).

**Animals.** *Pdx1-Cre, Pdx1-CreER*[TM42], *Lsd1*[flox37], and *Rosa26-eYFP*[43] mouse strains have been described previously. *Lsd1*[Δpan] mice were generated by crossing *Pdx1-Cre* and *Lsd1*[flox] mice. Conditional *Lsd1* knockouts were generated by crossing *Pdx1-CreER*[TM] and *Lsd1*[flox] mice. Tamoxifen (Sigma) was dissolved in corn oil (Sigma) at 10 mg/mL, and a single dose of 3.5 mg/40 g or 4.5 mg/40 g body weight was administered by intraperitoneal injection at embryonic day (e) 10.5 or e12.5,

respectively. Control mice were LSD1$^{+/+}$ littermates carrying the *Pdx1-Cre* or the *Pdx1-CreER*$^{TM}$ transgene. Midday on the day of vaginal plug appearance was considered embryonic day (e) 0.5. All animal experiments were approved by the Institutional Animal Care and Use Committees of the University of California, San Diego. The numbers of animals studied per genotype are indicated within each experiment. The sex of *Lsd1*$^{Δpan}$ embryos and newborn pups was not determined. *Lsd1*$^{Δpan}$ mice were not viable for more than 3–5 days after birth.

**Human tissue**. Human foetal pancreas donor tissue was obtained from the Birth Defects Research Laboratory of the University of Washington. Cadaveric adult pancreata used in this study were from non-diabetic donors and were acquired through the Network for Pancreatic Organ Donors with Diabetes (nPOD)[43]. Protein expression was analysed in nPOD donors: LSD1 and GCG in #6140 (38-year-old male); LSD1 and CHGA in #6160 (22-year-old male); LSD1 and SST in 6178 (25-year-old female); and LSD1, INS and GCG in 6179 (21-year-old female). All studies for use of foetal and adult human tissue were approved by the Institutional Review Board of the University of California, San Diego. Informed consent was obtained for use of human samples.

**Pancreatic endocrine differentiation of human embryonic stem cells (hESCs)**. Pancreatic differentiation was performed as previously described[5,12,14]. Briefly, a suspension-based culture format was used to differentiate cells in aggregate form. Undifferentiated aggregates of hESCs were formed by re-suspending dissociated cells in hESC maintenance medium at a concentration of $1 \times 10^6$ cells/mL and plating 5.5 mL per well of the cell suspension in 6-well ultra-low attachment plates (Costar). The cells were cultured overnight on an orbital rotator (Innova2000, New Brunswick Scientific) at 95 rpm. After 24 h the undifferentiated aggregates were washed once with RPMI medium and supplied with 5.5 mL of day 0 differentiation medium. Thereafter, cells were supplied with the fresh medium for the appropriate day of differentiation (see below). Cells were continually rotated at 95 rpm, or 105 rpm on days 4 through 8, and no media change was performed on day 10. Both RPMI (Mediatech) and DMEM High Glucose (HyClone) medium were supplemented with 1X GlutaMAX™ and 1% penicillin/streptomycin. Human activin A, mouse Wnt3a, human KGF, human noggin, and human EGF were purchased from R&D systems. Other added components included FBS (HyClone), B-27® supplement (Life Technologies), Insulin-Transferrin-Selenium (ITS; Life Technologies), TGFβ R1 kinase inhibitor IV (EMD Bioscience), KAAD-Cyclopamine (KC; Toronto Research Chemicals), and the retinoic receptor agonist TTNPB (RA; Sigma Aldrich). Day-specific differentiation media formulations were as follows: Days 0 and 1: RPMI + 0.2% (v/v) FBS, 100 ng/mL Activin, 50 ng/mL mouse Wnt3a, 1:5000 ITS. Days 1 and 2: RPMI + 0.2% (v/v) FBS, 100 ng/mL Activin, 1:5000 ITS. Days 2 and 3: RPMI + 0.2% (v/v) FBS, 2.5 mM TGFβ R1 kinase inhibitor IV, 25 ng/mL KGF, 1:1000 ITS. Days 3–5: RPMI + 0.2% (v/v) FBS, 25 ng/mL KGF, 1:1000 ITS. Days 5–8: DMEM + 0.5X B-27® Supplement (contains ~0.1 mg/L of retinol, all trans), 3 nM TTNPB, 0.25 mM KAAD-Cyclopamine, 50 ng/mL Noggin. Days 8–12: DMEM/B-27, 50 ng/mL KGF, 50 ng/mL EGF.

**LSD1 inhibition during pancreatic differentiation**. Early inhibition of LSD1 (LSD1i*$^{early}$*) was performed by addition of the irreversible LSD1 inhibitor tranylcypromine (TCP) to cell culture wells on days 7, 8 and 9 at a final concentration of 0.5 μM. Late inhibition of LSD1 (LSD1i*$^{late}$*) was performed by addition of TCP to cell culture wells on days 10, 11 and 12 at a final concentration of 0.5 μM. Additional LSD1 inhibitors used to perform LSD1i*$^{early}$* and LSD1i*$^{late}$* experiments were SP2509 (Selleck Chemicals), GSK-LSD1 (Sigma) and GSK2879552 (Chemietek); and experiments were performed in the same way as with TCP, at a final concentrations of 1 μM.

**Alteration of retinoic acid treatment during pancreatic differentiation**. Extended retinoic acid (RA) treatment (RA*$^{extended}$*) was performed by addition of the RA analogue TTNPB to cell culture wells on days 8 and 9 at a final concentration of 3 nM. Late RA treatment (RA*$^{late}$*) was performed by addition of TTNPB to cell culture wells on days 10 and 11 at a final concentration of 3 nM.

**Chromatin immunoprecipitation sequencing**. Chromatin Immunoprecipitation Sequencing (ChIP-seq) experiments for histone modifications were performed on day 10 with no TCP treatment or after TCP treatment (treatment on days 7, 8, and 9). ChIP-seq experiments for LSD1 were conducted on day 5 (GT stage) day 7 (PP1 stage) and day 10 (PP2 stage) without addition of TCP. ChIP-seq was performed using the ChIP-IT High-Sensitivity kit (Active Motif) according to the manufacturer's instructions. Briefly, for each cell stage and condition analysed, 5-10 ×10$^6$ cells were harvested and fixed for 15 min in an 11.1% formaldehyde solution. Cells were lysed and homogenised using a Dounce homogeniser and the lysate was sonicated in a Bioruptor® Plus (Diagenode), on high for 3 × 5 min (30 s on, 30 s off). Between 10 and 30 μg of the resulting sheared chromatin was used for each immunoprecipitation. Equal quantities of sheared chromatin from each sample were used for immunoprecipitations carried out at the same time. 4 μg of antibody were used for each ChIP-seq assay. Antibodies used were: rabbit anti-H3K27ac (Active Motif); rabbit anti-H3K4me1 (Abcam); rabbit anti-H3K4me2 (Millipore); rabbit anti-LSD1 (Abcam); goat anti-FOXA1 (Abcam); goat anti-

FOXA2 (Santa Cruz); goat anti-GATA4 (Santa Cruz); mouse anti-GATA6 (Santa Cruz); rabbit anti-HNF6 (Santa Cruz); and rabbit anti-RXRA (Santa Cruz). Chromatin was incubated with primary antibodies overnight at 4 °C on a rotator followed by incubation with Protein G agarose beads for 3 h at 4 °C on a rotator. Reversal of crosslinks and DNA purification were performed according to the ChIP-IT High-Sensitivity instructions, with the modification of incubation at 65 °C for 2–3 h, rather than at 80 °C for 2 h. Sequencing libraries were constructed using KAPA DNA Library Preparation Kits for Illumina® (Kapa Biosystems) and library sequencing was performed on a HiSeq 4000 System (Illumina®) with single-end reads of 50 base pairs (bp). Both library construction and sequencing were performed by the Institute for Genomic Medicine (IGM) core research facility at the University of California at San Diego (UCSD). Two replicates from independent hESC differentiations were generated for each ChIP-seq experiment, except for RXR where only one dataset was generated.

**Chromatin immunoprecipitation sequencing data analysis**. Single-end 50-bp ChIP-seq reads were mapped to the human genome consensus build (hg19/GRCh37) and visualised using the UCSC Genome Browser[44]. Bowtie 2, version 2.2.7[45] was used to map data to the genome and unmapped reads were discarded. SAMtools[46] was used to remove duplicate sequences and HOMER[47] was used to call peaks using default parameters and to generate tag density plots. Stage- and condition-matched input DNA controls were used as background when calling peaks. The BEDtools[48] suite of programs was used to analyze whether certain peaks overlapped with other peaks or modified histone regions. Differential peak analysis using HOMER, with default parameters, identified enhancer dynamics between the PP1 to PP2 stage and classify LSD1-bound enhancers into the different enhancer groups (G1, G2 and G3). Each ChIP-seq analysis was performed with two biological replicates, except H3K4me2 at GT, for which pseudo-replicates were generated. The first replicate was analysed and correlated with the appropriate second replicate. Pearson correlations between replicates are listed in Supplementary Table 1.

**RNA Isolation and sequencing (RNA-seq) and qRT-PCR**. RNA was isolated from cell samples using the RNeasy® Micro Kit (Qiagen) according to the manufacturer instructions. For each cell stage and condition analysed between 0.1 and $1 \times 10^6$ cells were collected for RNA extraction. For qRT-PCR, cDNA synthesis was first performed using the iScript™ cDNA Synthesis Kit (Bio-Rad) and 500 ng of isolated RNA per reaction. qRT-PCR reactions were performed in triplicate with 10 ng of template cDNA per reaction using a CFX96™ Real-Time PCR Detection System and the iQ™ SYBR® Green Supermix (Bio-Rad). PCR of the TATA binding protein (TBP) coding sequence was used as an internal control and relative expression was quantified via double delta $C_T$ analysis. For RNA-seq, stranded, single-end sequencing libraries were constructed from isolated RNA using the TruSeq® Stranded mRNA Library Prep Kit (Illumina®) and library sequencing was performed on a HiSeq 4000 System (Illumina®) with single-end reads of 50-bp. Both library construction and sequencing were performed by the IGM core research facility at UCSD. A complete list of RT-qPCR primer sequences can be found in Supplementary Table 2.

**RNA-seq data analysis**. Single-end 50-bp reads were mapped to the human genome consensus build (hg19/GRCh37) using the Spliced Transcripts Alignment to a Reference (STAR) aligner[49]. Tag directories were constructed from STAR outputs and normalised gene expression (fragments per kilobase per million mapped reads; FPKM) for each sequence file was determined using HOMER[47]. HOMER was used to annotate all RefSeq genes with FPKM values and to invoke the R package DESeq2[50] for differential expression analysis. Each RNA-seq analysis was performed on at least two biological replicates with DESeq2, using the built-in option to account for batch effects.

**Assignment of enhancer target genes and motif enrichment analysis**. Enhancer target genes were assigned using BEDtools to identify transcription start sites (TSSs) located ±100 kb from LSD1-bound enhancers (groups G1, G2 and G3). HOMER[47] was used to identify transcription factor (TF) binding motifs enriched in the G1 enhancer group versus the G2 and G3 groups. G2 and G3 enhancer peak files were merged and set as the background. G1 enhancers associated with one or more genes with FPKM ≥ 1 at the PP1 stage were used for motif analysis. Analogous motif enrichment analysis was conducted at G2 and G3 enhancers versus the entire genome, excluding G1, G2, and G3 regions.

**Flavin adenine dinucleotide measurements**. Flavin adenine dinucleotide (FAD) on hESC-derived cells was isolated with the FAD Assay Kit (Abcam, ab204710). Cell lysates were deproteinated using perchloric acid and FAD was quantified using the colorimetric assay according to the manufacturer's instructions. The isolation procedure was performed in triplicate with one to two million cells on differentiation days 0, 2, 5, 10, and 13. FAD measurements were performed simultaneously and normalized to protein concentration determined by the Micro BCA Protein Assay Kit (Thermo Scientific, 23235).

**Immunofluorescence analysis**. Cell aggregates derived from hESCs were allowed to settle in microcentrifuge tubes and washed twice with PBS before fixation with 4% paraformaldehyde (PFA) for 30 min at room temperature. Dissected e10.5–e13.5 mouse embryos, e15.5—postnatal day (P) 0 pancreata, and pancreata from 3-month-old mice were fixed in 4% PFA in PBS for 30 min, 3 h, and overnight, respectively. Fixed samples were washed with PBS and incubated overnight at 4 °C in 30% (w/v) sucrose in PBS. Samples were then loaded into disposable embedding molds (VWR), covered in Tissue-Tek® O.C.T. Sakura® Finetek compound (VWR) and flash frozen on dry ice to prepare frozen blocks. The blocks were sectioned at 10 μm and sections were placed on Superfrost Plus® (Thermo Fisher) microscope slides and washed with PBS for 10 min. Slide-mounted cell sections were permeabilised and blocked with blocking buffer, consisting of 0.15% (v/v) Triton X-100 (Sigma) and 1% (v/v) normal donkey serum (Jackson Immuno Research Laboratories) in PBS, for 1 h at room temperature. Slides were then incubated overnight at 4 °C with primary antibody solutions. The following day slides were washed five times with PBS and incubated for 1 h at room temperature with secondary antibody solutions. Cells were washed five times with PBS before coverslips were applied.

All antibodies were diluted in blocking buffer at the ratios indicated below. Primary antibodies used were: goat anti-carboxypeptidase A1 (Cpa1) (1:1000 dilution, R&D systems); goat anti-ghrelin (Ghrl) (1:300 dilution, Santa Cruz); goat anti-glucagon (Gcg) (1:1000 dilution, Santa Cruz); goat anti-osteopontin (Opn) (1:300 dilution, R&D systems); guinea pig anti-Ngn3 (1:1000,[51]); goat anti-PDX1 (1:500 dilution, Abcam); guinea pig anti-insulin (INS) (1:500 dilution, Dako); mouse anti-glucagon (GCG) (1:500 dilution, Sigma); rabbit anti-somatostatin (SST) (1:500 dilution, Dako); mouse anti-NKX6.1 (1:300 dilution, Developmental Studies Hybridoma Bank); rabbit anti-amylase (Amy) (1:500 dilution, Sigma); rabbit anti-chromogranin A (ChgA) (1:1000 dilution, Dako); rabbit anti-LSD1 (1:500 dilution, Abcam); rabbit anti-Phospo-Histone3 (Ser10) (pHH3) (1:1000 dilution, Cell Signaling); rabbit anti-polypeptide Y (Ppy) (1:1000 dilution, Dako); rabbit anti-Ptf1a (1:500 dilution, BCBC); rabbit anti-SOX9 (1:1000 dilution, Millipore); rat anti-E-cadherin (Cdh1) (1:300 dilution, Sigma); rat anti-EpCAM (1:100, DSHB # G8.8); sheep anti-NGN3 (1:300, R&D Systems). Secondary antibodies against sheep, rabbit, goat, mouse, rat, and guinea pig were Alexa488-, Cy3- and Cy5-conjugated donkey antibodies and were used at dilutions of 1:1000, 1:2000, and 1:250, respectively (Jackson Immuno Research Laboratories). Cell nuclei were stained with Hoechst 33342 (1:3000, Invitrogen). TUNEL staining was performed using the ApopTag® Plus Peroxidase In Situ Apoptosis Kit (Millipore). Representative images were obtained with a Zeiss Axio-Observer-Z1 microscope equipped with a Zeiss ApoTome and AxioCam digital camera. Figures were prepared in Adobe Creative Suite 5.

**Flow cytometry analysis**. Cell aggregates derived from hESCs were allowed to settle in microcentrifuge tubes and washed with PBS. Cell aggregates were incubated with Accutase® at room temperature until a single-cell suspension was obtained. Cells were washed with 1 mL ice-cold flow buffer comprised of 0.2% BSA in PBS and centrifuged at 200 × g for 5 min. BD Cytofix/Cytoperm™ Plus Fixation/Permeabilization Solution Kit was used to fix and stain cells for flow cytometry according to the manufacturer's instructions. Briefly, cell pellets were re-suspended in ice-cold BD Fixation/Permeabilization solution (300 μL per microcentrifuge tube). Cells were incubated for 20 min at 4 °C. Cells were washed twice with 1 mL ice-cold 1X BD Perm/Wash™ Buffer and centrifuged at 10 °C and 200 × g for 5 min. Cells were re-suspended in 50 μL ice-cold 1X BD Perm/Wash™ Buffer containing diluted antibodies, for each staining performed. Cells were incubated at 4 °C in the dark for 1 h. Cells were washed with 1.25 mL ice-cold 1X BD Wash Buffer and centrifuged at 200 × g for 5 min. Cell pellets were re-suspended in 300 μL ice-cold flow buffer and analysed in a FACSCanto™ (BD Biosciences). Antibodies used for flow cytometry: PE-conjugated anti-PDX1 (1:20 dilution, BD Biosciences); AlexaFluor® 647-conjugated anti-NKX6.1 (1:20 dilution, BD Biosciences); PE-conjugated anti-INS (1:50 dilution, BD Biosciences). Flow cytometry data was processed using FlowJo v10 software.

**Generation of LSD1 shRNA lentiviruses**. To generate shRNA expression vectors, shRNA guide sequences were placed under the control of the human U6 pol III promoter in the pLKO.1 backbone (Addgene, plasmid #10878). Small hairpin sequences are listed in Supplementary Table 3.

High-titer lentiviral supernatants were generated by co-transfection of the shRNA expression vector and the lentiviral packaging construct into HEK293T cells as described[12]. Briefly, shRNA expression vectors were co-transfected with the pCMV-R8.74 (Addgene, #22036) and pMD2.G (Addgene, #12259) expression plasmids into HEK293T cells using a 1 mg/ml PEI solution (Polysciences). Lentiviral supernatants were collected at 48 h and 72 h after transfection. Lentiviruses were concentrated by ultracentrifugation for 120 min at 19,500 rpm using a Beckman SW28 ultracentrifuge rotor at 4 °C.

**Transduction of hESC endodermal lineage intermediates**. Differentiation toward the pancreatic progenitor cell stage was initiated as described. At day 6 of differentiation, cells were dissociated with Accutase, washed in PBS + 0.02% BSA and counted. Cells were then distributed onto a 6 well plate at a density of 5 million

cells per well. Concentrated lentivirus was then added at 1 μL/mL media, as well as 8 μg/mL polybrene, 10 μM Rock Inhibitor, and 5 units/mL heparin. Cells were then re-aggregated at 100 rpm. After 6 h, viral media was replaced with fresh day 6 differentiation media. Differentiation was then continued as described, and cells were collected for analysis at day 13.

**RNAscope**. Mouse embryos at e12.5 were fixed in 4% PFA at 4 °C overnight, embedded in OCT (Sakura Finetek), frozen, and sectioned at 10 μm. Serial sections were prepared as described in the "Immunofluorescence analysis" section. The expression of mouse *HoxA1* transcripts was detected using the RNAscope Probe-Mm-Hoxa1 #542391, Multiplex Fluorescent Reagent Kit v2 (#323100, Advanced Cell Diagnostics) and Opal 570 Reagent Pack (Akoya Biosciences #FP1488001KT) according to the manufacturers' recommendations, with the following specifics: target retrieval was omitted, slides received a 30 min treatment with Protease IV and Opal 570 fluorophore and DAPI was diluted at 1:1500. Images were captured with a Zeiss Apotome microscope.

**Gene ontology**. Gene ontology analysis was performed using Metascape (http://metascape.ncibi.org) with the default parameters.

**Principle component analysis**. For all samples, FPMKs for total transcriptome were calculated as described above. Genes were then filtered for FPKMs greater than or equal to one, and genes showing the top 25% median absolute deviation (MAD) were selected. Based on these values, PCA plots were generated using the PRComp package in R.

**Morphometric analysis and cell counting**. At e12.5, every pancreas section was analysed from a minimum of three embryos per genotype, while every fifth pancreas section was analysed at e15.5. For determination of total pancreatic epithelial area, E-cadherin+ area per section was measured using ZEN Digital Imaging for Light Microscopy software (ZEISS), which was calibrated to calculate values in μm$^2$. The number of marker+ cells was determined by counting every marker+/DAPI+ cell in each section. The number of marker+ cells per section was subsequently divided by the total epithelial area of the section and expressed as marker+ cells/μm$^2$. All values are shown as mean ± standard error of the mean (SEM); $p$-values calculated using unpaired Student's $t$-test; $p < 0.05$ was considered significant.

**Experimental comparisons**. All experiments were independently repeated at least twice. Results are shown as mean ± SEM. Statistical analyses were conducted using GraphPad Prism 6 and R.

**Differential gene expression analysis**. The DESeq2 Bioconductor package for R was used to calculate gene expression changes. Adjusted $p$-values < 0.05 and $\log_2$(fold-change) ≥1.5 were considered significantly different.

**Permutation-based significance**. A random sampling approach (10,000 iterations) was used to obtain null distributions for enrichment analyses, in order to obtain $p$-values. Null distributions for enrichment of enhancers for gene sets were obtained by randomly shuffling enhancer regions using BEDtools and overlapping with nearby genes. $P$-values < 0.05 were considered significant.

**Reporting summary**. Further information on research design is available in the Nature Research Reporting Summary linked to this article.

## Data availability

The authors declare that all data supporting the findings of this study are available within the article and its Supplementary Information files or from the corresponding author upon reasonable request. The raw data reported in this manuscript for the ChIP-seq and RNA-seq data have been deposited in the GEO database under accession code GSE104840. The accession code for previously reported H3K4me1 and H3K27ac ChIP-seq data is GSE54471. The accession code for previously reported RNA-seq data is E-MTAB-1086. The source data underlying Figs. 1c, 4h, 6c, and Supplementary Figs. 1b, c, h, 2c, 3c, 6b, d, 7b, e, f, and 8d, e are provided as a Source Data file.

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

## Acknowledgements

We thank FenFen Liu, Nancy Rosenblatt, Karen V. Navarro, and Thomas Harper for experimental assistance, Yinghui Sui for help with data analysis, and the Sander laboratory for helpful discussions and comments on the manuscript. We are grateful to Tung Nguyen, Rizi Ai and Wei Wang for help with data analysis at early stages of this project and Sven Heinz for advice with ChIP-seq experiments. We acknowledge support of the UCSD Human Embryonic Stem Cell Core for cell sorting and the UCSD IGM Genomic Center (supported by P30 DK064391) for library preparation and sequencing. This work was supported by a California Institute for Regenerative Medicine (CIRM) training grant (N.A.P.), Juvenile Diabetes Research Foundation postdoctoral fellowships (H.P.S., A.W.), National Institutes of Health (NIH) grant T32 GM008666 (R.G), and by NIH grants DK089567 and DK068471, and CIRM grant RB5-07236 (M.S.).

## Author contributions

N.K.V., N.A.P., A.W., and M.S. conceived the project and designed the experiments. N.K.V., N.A.P., R.J.G., I.M., A.R.H., H.P.S., N.W., and Jinzhao W. performed experiments. N.K.V., R.J.G. and C.W.B. analysed sequencing data. Jianxun W. and M.G.R. provided mice. H.P.S., M.W. and U.S.J. provided technical expertise and critical suggestions for experimental design. N.K.V., N.A.P., R.J.G., A.W., H.P.S., and M.S. interpreted data. N.K.V., N.A.P., R.J.G., and M.S. wrote the paper.

## Competing interests

The authors declare no competing interests.
