## [Peer Review File · Nature Communications]

Reviewers' Comments:

Reviewer #1:

Remarks to the Author:

The work by Vinckier et al. examines the role of LSD1 in enhancer decommissioning during endodermal (gut/pancreas) embryonic development. The authors show that LSD1 binding to enhancers induces demethylation of H3K4 and promotes enhancer decommissioning. In this way, LSD1-mediated enhancer silencing acts as an epigenetic intrinsic mechanism to switch off the transcriptional response to an inductive developmental signal. This mechanism could explain how signaling cues are interpreted in developing tissues and how they are temporally limited to defined windows of competence for developmental transitions.

The authors use hESC and mouse knockouts to show that early inhibition or deletion of LSD1 abolish the later production of endocrine pancreatic cells. Lack of LSD1 results in persistence of H3K4me1 and me2 in enhancers that remained poised for activation. Genomic analysis of activated genes in LSD1 early inhibition (G1 group at enhancers) indicated the involvement of retinoic acid signaling. Persistence of RA signaling upon early LSD1 inhibition seems to prevent further endocrine differentiation.

The manuscript is very well written and data are clearly explained. The questions are highly relevant to a wide audience of developmental biologists and provide a framework to understand longstanding questions in the field. There are however a few criticisms and suggestions that should be addressed.

Major points

1. The B27 supplement the authors use in their experiments usually contains retinoic acid (RA)/vitaminA at variable concentrations, usually around 1ug/ml. Can the authors please clarify whether the B27 they use in their culture at days 8-12 contains retinoic acid or is without it? If it contains RA, can the authors perhaps rephrase the RA extension/persistence concept with the more appropriate dampening of response to RA, which is also supported by the data using RA late activation after early LSD1 inhibition?
2. N number is equal 2 for most experiments. We understand that the phenotype is quite clear and strong, so it might not need a third experiment to reach significance, but it would be good to know that the authors have repeated it multiple times and they are confident about the (non) variability of their assays (eg hESC differentiation protocol, area of the embryonic epithelium?)
3. In figure 5, if the area of the epithelium is not changed in LSD1 knockouts and yet the endocrine cells/islets (which should be 2-6%) are lost, is there any fate switch? What happens to the cells that do not differentiate to the endocrine lineage? Do they remain stuck at an earlier stage? The authors could look at the expression Ngn3 to comment on where in the lineage cells remain stuck.
4. What about the role of LSD1 on G2 and G3 groups? For example, G3 group has less acetylation already in PP1, which is a prerequisite for LSD1 demethylase activity, and yet bound LSD1 does not remove methyl groups (methylation is actually increased)? Why?
5. Figure 4 looks at upregulated genes and their relationship to RA signaling. If similar analysis is done with the down-regulated genes, is no relationship observed?
6. RA late treatment seems to give rise to multihormonal cells (fig 5b)? This deserves comment. The authors could try to treat pancreatic explants with RA early and late and look at endocrine cells to see both whether they can recapitulate LSD1 knockout data but also whether RA treatment

at later stages results in polyhormonal cells. This could be really important for understanding the mechanisms that segregate endocrine subtypes and might be relevant for the generation of mono-hormonal insulin-producing beta cells for diabetes.

7. Fig 6f: why do you have so many LSD1 positive cells in LSD1Delta-early (around but also far from insulin+ cells)? Is this picture representative?

Minor points

1. It would be helpful to add in figure 1 the days to reach the different intermediates
2. Is LSD1 ubiquitous? What about other tissues in which RA plays an important and biphasic role? For ex in the nervous system, RA is first involved in posteriorization and then it is essential for neuronal differentiation. The authors should add something about this in the discussion.
3. If RXR ChIPseq has been undertaken and is referred to, the data about location of peaks etc. should be made available.

Reviewer #2:

Remarks to the Author:

NCOMMS-18-30737 - LSD1-mediated enhancer silencing enables endocrine cell development through attenuation of retinoic acid signalling

Nicholas K. Vinckier, Nisha A. Patel , Allen Wang , Jinzhao Wang , Hung-Ping Shih, Jianxun Wang , Ulupi S. Jhala , Michael G. Rosenfeld , Christopher W. Benner and Maïke Sander

In this manuscript the authors, using a well-defined stepwise differentiation program of hECS into pancreatic endocrine cells along with relevant genomic approaches, suggest that LSD1 affects the epigenetics landscape of the differentiation program in a defined window and demonstrate that LSD1-mediated enhancer decommissioning affects retinoic acid (RA)-induced expression. The authors also show that, in the absence of LSD1, RA-dependent enhancers remain active despite the absence of RA, with a consequent failure to progress into the differentiation program. Collectively, their findings indicate that LSD1 functions at a defined window of the differentiation program, and loss of LSD1 alters epigenetics landscape affecting RA signaling. The findings reported are strongly indicative of the role of LSD1 in the appropriate control of endocrine program and highlight a novel time-dependent function of LSD1 in endocrine differentiation. I share with the authors the possibility that their findings might be broadly relevant.

Overall, the authors have made potentially interesting findings but, some data presented in the MS are over-interpreted and there are several areas where it falls somewhat short. Some additional experiments as well as a more accurate presentation of the data will definitely improve the significance of their observations.

Specific comments

1. A compounding issue is that the authors consider the use of drugs such as the irreversible TCP and the reversible SP2509 (along with GSK-LSD1 and GSK2879552) equivalent to LSD1 knock-out and this is not fully correct. These drugs, especially TCP (used to generate most of the data), also inhibit the activity of other proteins. This point raises the question of the selectivity of the LSD1 inhibitors since TCP in addition to inhibit monoamine oxidases MAOA/B, also affects the LSD1 paralog LSD2. I appreciate the in vivo data presented in Figure 6 supporting the specific role of

LSD1 in endocrine cell formation, however, they are insufficient to validate all data obtained in the in vitro studies using pharmacological inhibition of LSD1. It is pertinent to consider that like LSD1, LSD2/KDM1B is a FAD-dependent enzyme histone demethylase and it participates to gene transcription at least in part through control of transcription elongation control. I would urge the authors to determine the specificity of LSD1 inhibition at PP1-PP2 stage by CRISPR-Cas9 editing (or similar specific LSD1 genetic ablation). Alternatively, if LSD2 is expressed in hESCs, (likely to be the case), LSD2 Chip-seqs in the presence and absence of TCP would help to segregate the relative contribution of both KDM1A and KDM1B at PP1 to PP2 transition.

2. ChIP-seqs for LSD1 were carried out at PP1 stage with or without LSD1 inhibitor (I assume TCP??). For how long the cells were treated with TCP before chromatin isolation? Specifically, I wonder whether the chromatins from treated and untreated cells were prepared at the same time of differentiation protocol. As it stands the LSD1 inhibitor protocol is unclear.

3. Previous work in mouse ESCs (Whyte et al. 2012) reported LSD1 signals at core promoter regions. In addition, several reports indicate the presence of LSD1 signals at TSS. From the presented data it is not clear how many of the 15,084 LSD1 peaks are at TSS? And.... if there are examples where LSD1 is present in both distal and core TSS, or only at distal, or only at TSS. Please provide these informations.

4. The authors state that they:..... isolated enhancers that are active at PP1 and/or PP2 and also bound by LSD1 at PP1,These putative enhancers were divided in 3 groups G1, G2 and G3, on the basis of the relative abundance of H3K27ac. I assume that the authors imply that the distal LSD1-bound regions (enhancers) are also co-occupy by hESC master transcription factors such Oct4, Sox2, Myc and Nanog and the Mediator coactivator. I reckon that this assumption must be validated. It will be sufficient to demonstrate by ChIP data that a specific distal LSD1 binding region (such as HOXA1 shown in Fig 2 E) is also co-occupy by a dedicate TF or Mediator factor.

5. LSD1 is a FAD-dependent enzyme. Is the FAD level constant throughout the hECS differentiation program toward pancreatic endocrine cell lineage? The rate-limiting role for FAD in LSD1-mediated effects has been recently documented by Yang et al. EMBO J. 2017, 36: 1011–1028, doi:10.15252/embj.201694408. Reduction of FAD levels at PP2 stage will provide a plausible explanation of the temporal window of LSD1 function.

6. In fig 1 the effects of LSD1 (both early and late) at EN stage is revealed only by flow cytometry. I suggest that the authors validate the phenotype also by qRT-PCR, as shown in panel A of Supplementary Fig 5.

7. Provide qRT-PCR of LSD1 or WB in Supp. 1a (no FPKM). It is not clear if the LSD1 wave of expression (low in DE, high in GT) is significant. Similarly, data from Fig. 4 panel H, and Supplementary Fig 6 E, should be validated by qRT-PCR.

8. Legend Supp. Fig. 3. CYP26A1 is duplicated.

9. To provide sufficient details for the work to be reproduced successfully, the authors should add details of the pharmacological conditions used.

10. The description of figures is often rather cryptic. The figure legends need to be specified so that one can understand what was done. As it stands too much guessing is necessary.

Reviewer #3:

Remarks to the Author:

This manuscript provides evidence that the histone demethylase LSD1 regulates the epigenetic

status of developmental enhancers to control how those enhancers respond to extracellular signals, namely retinoic acid during pancreas specification. While LSD1 is known to “decommission” enhancers in the context of cell signaling the current study suggests that this provides a mechanistic basis for developmental competence and how the duration of transcriptional responses to extracellular signals are modulated and reset during lineage progression. This is an important area of work that in principle the findings represents an important advance in the field. However, there are a few important issues that need to be addressed to support the model.

1. What recruits LSD1 to enhancers and when is it first there. If the model is correct then LSD1 should not be at G1 enhancers at the gut tube stage. If it is at there, then it should prevent those enhancers from becoming active in response to RA signaling. The authors really need to show LSD1 binding at the GT-stage to address this point as it is a key aspect of the model.

2. The idea that LSD1 regulates competence of an enhancer to respond to RA-RxR activity is attractive, but the data that the authors show is mostly correlative. If the model is correct then ectopic LSD1 should prevent (or attenuate) RA from activating these enhancers.

3. The data and interpretation in Fig. 2 d-f is confusing. The model proposes that LSD1 demethylates after the enhancer after RA is withdrawn from the culture and that in the LSD1i demethylation doesn't happen and RA-responsive genes expression persists. But if this is true then I would expect an increase in K27ac in the LSD1i to accompany the persistent gene expression. How do you get persistent expression in the absence of K27Ac?

4. The evidence that the modest ~2-fold increase in me2 on G1 peaks can account for the persistent expression is not very strong. These two observations might be true-true and unrelated. An alternative explanation is that LSD1 is doing something else, independent of K4me2 levels to regulate expression.

5. In Fig. 4h it would be good to also show the expression of these key LSD1/RA-responsive genes at PP1 so that the reader has a clear understanding of what this elevated expression at PP2 is relative to that seen normally at the PP1 stage. The elevation could be very modest only 10% of PP1 levels. Fig. 3d is a good stat but the authors should show the expression of the key LSD1/RA-targets. Similarly in Fig. 5 it would be good to show the RNA-seq/RT-PCR data in a histogram to more clearly show the expression of key LSD1/RA-responsive genes at PP1, PP2 and EN +/-late RA to be able to compare the level of expression. Are the LSD1/RA-regulated genes really not activated at all when RA is added late (it is hard to extract that info from the current figure). When they are activated in the LSD1i+RALate condition what is the level of expression relative to that normally seen at PP1.

6. If the LSD1i or RA extended cells they are stuck at a progenitor state then PCA analysis of the RNA-seq might show that the PP2-LSD1i or PP2-extended RA are more similar to PP1 cells.

7. For the mouse genetics any evidence of elevated RA-target gene (Hox) expression in the LSD1 KO?

Minor points

1. Need to report the antibodies used for ChIP

2. While the enrichment analysis of motifs is okay, the authors should show the motif analysis of LSD1 ChIP peaks from G2, G3 and the rest of the genome other than G1-3. Perhaps RxR is just a common feature of all of these.

3. The authors definition of decommissioned and deactivated enhancers changes a bit at different points in the paper. G1 enhancers are selected as "decommissioned" based on loss of K27ac. But then they classify decommissioned based on the removal of me2, in the context of no change in K27ac.

Reviewer #4:

Remarks to the Author:

Summary:

In this manuscript the authors first show that endocrine cell development in differentiating hESC cultures requires LSD1 activity during a limited time window when into the first recognizable pancreas progenitors (PP1 cells) differentiate into late pancreas progenitors (PP2 cells), but not during differentiation of PP2 cells into the endocrine lineage (EN cells).

The authors then conducted LSD1 ChIP-seq to identify potential target genes. Analyses of the LSD1 ChIP-seq dataset together with previously published ChIP-seq data sets for H3K27Ac and H3K4 Me1/2 identified an LSD1-regulated set of enhancers (G1) that is activated when gut tube (GT) progenitors evolve into PP1 cells and deactivated (deacetylated) and decommissioned (demethylated) when these factors are withdrawn during the PP1 to transition PP2. The authors found that deacetylation of these enhancers occurred largely independent of LSD1, but that LSD1 was required for demethylation. Comparison to RNA-seq data from LSD1-inhibited cells suggested that genes up-regulated after LSD1 inhibition were direct LSD1 target genes, whereas down-regulated genes were not directly regulated by LSD1. The authors then found an enrichment for genes involved in RA signaling among G1 enhancers.

Transcription factor binding motif enrichment analysis showed significant enrichment of the motif for the RAR/RXR heterodimer at G1 enhancers and RXR ChIP-seq analysis indicated binding of RAR/RXR receptors to about one third of the G1 enhancers. The authors then show that extending the RA exposure into the PP1 to PP2 transition mimicked the phenotype seen upon LSD1 inhibition, while progenitor cell markers were unaffected. RNA-seq analysis was then used to identify genes dysregulated as a result of prolonged RA exposure and genes up-regulated by LSD1 inhibition were enriched among the genes also up-regulated after extended RA exposure. Similar to the LSD1 inhibition, this effect of prolonged RA treatment was not seen at the later PP2 to EN transition. Interestingly, prior LSD1 inhibition led to enhanced induction of RA responsive genes, suggesting that LSD1 dampens future RA responsiveness in cells that experienced prior RA exposure.

Lastly, the authors demonstrate that pancreatic *Lsd1* deletion in vivo in mice results in a strong reduction in endocrine cells and that this *Lsd1* dependency is restricted to a narrow time window between E10.5 and E12.5.

Critique:

This is a clearly written paper with some interesting and novel findings. Generally, the experiments are well performed and many of the conclusions drawn are justified. However, the main conclusion is not supported by the data in their present form. Thus, I would like to see the following major issues addressed before recommending publication:

1) The motif search done by the authors, which revealed the RA-response element being enriched is not particularly impressive given that Table S6 shows that only 1.14% of the LSD1 target sequences had this motif. Adding the DR1 motif (line 20 in Table S6) at 2.4% only slightly improves this. It is thus surprising that nearly one third of the G1 enhancers showed RXR binding, but without access to the ChIP-seq data, one cannot assess the significance of this observation. The GEO data set should therefore be made available to the reviewers.

2) The phenotypic similarity between LSD1 inhibition and prolonged RA exposure is, albeit

interesting, mostly correlative in nature and none of the experiments directly test whether the effect of LSD1 inhibition is solely caused by a failure to decommission RA-responsive enhancers and thus prolonged activity of the genes regulated by those enhancers.

Thus, the link between LSD1 and RA signaling is somewhat tenuous. To bolster their conclusions the authors should test whether inhibition of RA signaling, for example with an inverse agonist that prevents activation even from "open" enhancers, can rescue the effect of LSD1 inhibition. This should be done both in the hESC cultures and in explants from Lsd1-deficient embryonic mouse pancreas.

3) Endocrine cells differentiate from pancreatic progenitor cells via a NEUROG3 expressing precursor stage, yet nothing is mentioned about this in the paper. In the discussion the authors refer to "the endocrine-specified pancreatic progenitor cell stage" but it is unclear what they mean by this. Do the authors mean NEUROG3+ cells here?

More importantly, the authors make no attempt to decipher whether the effect of LSD1 inhibition or extended RA treatment on EN cell differentiation is executed by preventing induction of NGN3 or preventing the differentiation of NGN3+ cells into EN cells. The tools for making this distinction are readily available and have previously been used by the authors. This is of major importance for the field and for the usefulness of these findings in the translation to hESC differentiation protocols. Thus, this omission should be remedied.

Minor issues:

Antibodies used for ChIP-seq are not listed in the methods section.

We thank the reviewers for their time and effort in reviewing our manuscript. Please find below a summary of the revision as well as a point-by-point response to each of the reviewers.

We are pleased that our study was well received as evidenced by comments such as “the study is highly relevant to a wide audience (reviewers 1 and 2)”, “the findings represent an important advance in the field (reviewer 3)”, “experiments are well performed and many of the conclusions justified (reviewer 4)”, and the manuscript is “well written and explained (reviewers 2 and 4)”. We appreciate the thoughtful comments we received and have addressed the reviewers’ concerns in the present manuscript. While the reviewers highlighted the novelty and importance of identifying LSD1 as a time-dependent regulator of pancreatic endocrine cell differentiation, they also asked us to further define which step in endocrine cell differentiation is affected by loss of LSD1, and to strengthen the link between LSD1 and the regulation of RA signaling.

We have taken great effort to fully address each of the reviewers’ concerns and have revised the manuscript extensively to clarify existing data and accommodate new data. We believe the manuscript now provides stronger evidence for our conclusions and clearly demonstrates a conserved role for LSD1 in pancreatic endocrine differentiation and the regulation of RA signaling. Key additions and revisions include:

Additional insight into how LSD1 controls endocrine cell differentiation

Reviewers 1 and 4 asked us to further define the mechanism by which LSD1 regulates endocrine cell differentiation and to specifically analyze whether endocrine lineage specification is affected. In the revised manuscript we show that expression of the endocrine progenitor marker NGN3 is not negatively affected in (i) *Lsd1*^{Δpan} mouse embryos (**Fig. 6b,c**), (ii) hESC-derived pancreatic progenitors treated with the LSD1 inhibitor (**Suppl. Fig. 1g,h**), and (iii) hESC-derived pancreatic progenitors exposed to extended RA (**Suppl. Fig. 6a,b**). The similarity in phenotype between the models further strengthens the applicability of the hESC system for identifying in vivo-relevant mechanisms. Furthermore, to strengthen mechanistic insights (reviewers 2 and 3), we added data showing that early pancreatic TFs, such as FOXA1/2, GATA4/6, and HNF6 recruit LSD1 to pancreatic enhancers, including enhancers of RA-responsive genes (**Suppl. Fig. 4b,c**).

Further evidence for link between LSD1 inhibition and regulation of RA-responsive genes

To strengthen the link between LSD1 and the regulation of RA signaling, we analyzed *HoxA1* expression in pancreas from *Lsd1*^{Δpan} mouse embryos (reviewer 3). As seen in the hESC system, we found *HoxA1* to be more highly expressed after *Lsd1* deletion in vivo (**Fig. 6g**). Furthermore, our data showing that NGN3 expression is maintained after (i) *Lsd1* deletion in mice, (ii), LSD1 inhibition in the hESC system, and (iii) prolonged RA exposure provides further evidence for similarity in phenotype across models, and therefore strengthens our model that prolonged RA signaling is causative for the endocrine differentiation block after LSD1 inhibition or deletion. Finally, we conducted additional data analyses which support the notion that LSD1-occupied RA-responsive enhancers are linked to the phenotype.

Validation of LSD1 inhibitor results through shRNA-mediated loss of function in hESCs

To further ascertain that effects by the LSD1 inhibitors can be explained by LSD1 function (reviewer 3), we show that transduction of hESC-derived pancreatic progenitors with two different shRNAs against *LSD1* phenocopies the effect of LSD1 inhibitors (**Suppl. Fig. 3**).

Below, we listed the reviewers’ comments in black and our responses in blue. Revised text in the manuscript is shown in blue.

Reviewer #1

The work by Vinckier et al. examines the role of LSD1 in enhancer decommissioning during endodermal (gut/pancreas) embryonic development. The authors show that LSD1 binding to enhancers induces demethylation of H3K4 and promotes enhancer decommissioning. In this way, LSD1-mediated enhancer silencing acts as an epigenetic intrinsic mechanism to switch off the transcriptional response to an inductive developmental signal. This mechanism could explain how signaling cues are interpreted in developing tissues and how they are temporally limited to defined windows of competence for developmental transitions.

The authors use hESC and mouse knockouts to show that early inhibition or deletion of LSD1 abolish the later production of endocrine pancreatic cells. Lack of LSD1 results in persistence of H3K4me1 and me2 in enhancers that remained poised for activation. Genomic analysis of activated genes in LSD1 early inhibition (G1 group at enhancers) indicated the involvement of retinoic acid signaling. Persistence of RA signaling upon early LSD1 inhibition seems to prevent further endocrine differentiation.

The manuscript is very well written and data are clearly explained. The questions are highly relevant to a wide audience of developmental biologists and provide a framework to understand longstanding questions in the field. There are however a few criticisms and suggestions that should be addressed.

Response:

We are grateful for the reviewer's appreciation of our work and for the recognition of its significance. Below, we addressed each point raised by this reviewer. Most notably, we added additional data on progenitor cell fate in the absence of LSD1 across all models and further analysis of how LSD1-regulated enhancers link to genes regulated by LSD1 and RA.

Major points

1. The B27 supplement the authors use in their experiments usually contains retinoic acid (RA)/vitaminA at variable concentrations, usually around 1ug/ml. Can the authors please clarify whether the B27 they use in their culture at days 8-12 contains retinoic acid or is without it? If it contains RA, can the authors perhaps rephrase the RA extension/persistence concept with the more appropriate dampening of response to RA, which is also supported by the data using RA late activation after early LSD1 inhibition?

Response:

We thank the reviewer for the comment. The B27 supplement used in our experiments is indeed not entirely RA-free and contains ~0.1mg/L of retinol, all trans. We have amended the methods section to document this and have also rewritten the manuscript to account for the fact that we are not completely withdrawing RA in the controls.

Line 225: *"During the PP1 to PP2 transition, the only possible source for stimulation of the RA receptor (RAR) are traces of retinol in the B27 supplement."*

2. N number is equal 2 for most experiments. We understand that the phenotype is quite clear and strong, so it might not need a third experiment to reach significance, but it would be good to know that the authors have repeated it multiple times and they are confident about the (non) variability of their assays (eg hESC differentiation protocol, area of the embryonic epithelium?)?

Response:

We apologize that the presentation of our data suggested that experiments have only been conducted twice. This is not the case. The LSD1 inhibition experiment in the hESC system was performed no less than 10 times, and in each case, we observed a complete block in endocrine

cell differentiation when the inhibitor was added at the PP1 pancreatic progenitor cell stage, whereas no decrease in endocrine cell numbers was observed when the inhibitor was added at the PP2 stage. Experiments in which endocrine cell differentiation was assessed after extended RA treatment, or late RA treatment were conducted at least 3 times and again, effects on endocrine cell differentiation were consistent across experiments. Our main endpoint for repeat experiments was immunofluorescence staining for endocrine hormones – flow cytometry was used as a confirmatory assay and not conducted for all repeat differentiations. The absence of endocrine cells in *Lsd1*^{Δpan} mice was observed in at least 6 embryos in which several sections throughout the entire pancreas were analyzed. We amended all figure legends to clearly state how often each experiment was conducted. Furthermore, we validated the LSD1 inhibitor phenotype in hESCs via shRNA-mediated knockdown of *LSD1* in PP1 cells (**Suppl. Fig. 3**).

3. In figure 5, if the area of the epithelium is not changed in LSD1 knockouts and yet the endocrine cells/islets (which should be 2-6%) are lost, is there any fate switch? What happens to the cells that do not differentiate to the endocrine lineage? Do they remain stuck at an earlier stage? The authors could look at the expression Ngn3 to comment on where in the lineage cells remain stuck.

Response:

As suggested, we conducted Ngn3 staining on pancreatic sections of *Lsd1*^{Δpan} mouse embryos. As shown in revised **Fig. 6b,c**, Ngn3 expression was unaffected, suggesting that loss of *Lsd1* specifically affects the differentiation of endocrine progenitors but does not affect endocrine lineage commitment. To further strengthen the validity of our hESC model and the link between LSD1, RA signaling, and endocrine cell differentiation, we also conducted NGN3 staining in hESC-derived pancreatic progenitors treated with the LSD1 inhibitor or exposed to extended RA. As in *Lsd1*^{Δpan} mice, NGN3 protein expression was not affected (**Suppl. Fig. 1g**; **Suppl. Fig. 6a**). At the mRNA level, we found *NGN3* expression to trend higher in both the LSD1 inhibitor-treated (**Suppl. Fig. 1h**) and RA^{extended} cultures (**Suppl. Fig. 6b**) when compared to controls. By showing further similarity in phenotype after LSD1 inhibitor and prolonged RA treatment, these new results strengthen the overall conclusions of our manuscript.

Furthermore, to address the question of whether *Lsd1*-deficient cells undergo a cell fate switch, we crossed a *Ngn3-YFP* allele into the *Pdx1-CreER;Lsd1*^{fllox/fllox} background to follow *Ngn3*-expressing cells short-term. YFP is stable for > 24 hours and therefore serves as a short-term lineage tracer during embryogenesis (PMID: 15297605). As previously shown (PMID: 15297605), we found YFP⁺ cells in controls (*Pdx1-CreER;Lsd1*^{fllox/+}; *Ngn3-YFP* mice injected with tamoxifen at e10.5) to be contiguous with and adjacent to the Sox9⁺ progenitor cell epithelium (**Reviewer Fig. 1a**). In the *Lsd1*-deficient background (*Pdx1-CreER;Lsd1*^{fllox/fllox}; *Ngn3-YFP* mice injected with tamoxifen at e10.5), YFP⁺ cells were detectable, which confirms findings from the *Ngn3* immunostaining. The pattern of YFP⁺ cells in *Lsd1*-deficient embryos resembled the one in control embryos with YFP⁺ cells being close to Sox9⁺ ducts (**Reviewer Fig. 1b**). This indicates that endocrine-fated progenitors do not adopt an acinar cell fate in the absence of *Lsd1* but instead revert to a ductal fate. While we included the *Ngn3* staining into the manuscript (**Fig. 6b,c**), we felt that the short-term lineage tracing might be too much detail with no immediate relevance to the overall conclusions of our study. However, upon the reviewer's request, we will also include these data into the final manuscript.

Reviewer Figure 1. Short-term lineage tracing of endocrine-fated progenitors in *Lsd1*-deficient mice reveals cell allocation to the ductal epithelium.

(a) *Pdx1-CreER;Lsd1^{flox/+};Ngn3-YFP* mice injected with tamoxifen at e10.5 served as controls. At e15.5, YFP is expressed from the *Ngn3-YFP* transgene and marks endocrine-fated progenitor cells, which localize within and adjacent to the Sox9⁺ ductal progenitor cell epithelium.

(b) *Lsd1* was inactivated by injecting *Pdx1-CreER;Lsd1^{flox/flox};Ngn3-YFP* mice with tamoxifen at e10.5. YFP expression shows that the endocrine lineage program is initiated independent of *Lsd1*. YFP⁺ cells are localized within and adjacent to the Sox9⁺ ductal progenitor cell epithelium in a pattern similar to control mice.

4. What about the role of LSD1 on G2 and G3 groups? For example, G3 group has less acetylation already in PP1, which is a prerequisite for LSD1 demethylase activity, and yet bound LSD1 does not remove methyl groups (methylation is actually increased)? Why?

Response:

We currently do not know how exactly LSD1 regulates chromatin state at G2 and G3 enhancers. G3 enhancers gain not only H3K4me2/1, but also H3K27ac during the PP1 to PP2 transition. It has been shown that histone acetylation overrides LSD1 activity and that in the presence of HATs and acetylated histones, LSD1 does not demethylate its substrates (PMID: 22297846). Therefore, it is not surprising that G3 enhancers gain H3K4me2/me1 while gaining H3K27ac during the PP1 to PP2 transition despite presence of LSD1. Mechanistically, LSD1 itself has been shown to be the substrate of HATs and HDACs, and LSD1 acetylation dampens LSD1 recruitment to nucleosomes and/or LSD1 stability (PMID: 27292636; 27212032; 27977115). Thus, emerging evidence suggests that LSD1 can only demethylate enhancers when HATs are not present, and HATs are likely recruited to G3 enhancers during the PP1 to PP2 transition.

Of relevance to this study, since G3 enhancers gain H3K27ac after RA is removed from the culture medium, and G2 enhancers do not show a significant change in H3K27ac levels in response to RA, acetylation of G2 and G3 enhancers must be mediated by an RA-independent mechanism. Our rationale for focusing on G1 enhancers is that LSD1 inhibition had the most significant effect on chromatin state at G1 enhancers (**Fig. 2d** and **Suppl. Fig. 4c**).

Furthermore, G1 enhancers associated more closely with LSD1-regulated genes than G2 and G3 enhancers (**Fig. 3b**). Since the focus of this manuscript is the role of LSD1 in endocrine cell differentiation and its relationship with the regulation of RA-responsive genes, we would respectfully argue that it is beyond the scope of this manuscript to determine exactly why methylation can occur in the presence of LSD1 at G3 enhancers.

5. Figure 4 looks at upregulated genes and their relationship to RA signaling. If similar analysis is done with the down-regulated genes, is no relationship observed?

Response:

We performed the proposed analysis and found that genes down-regulated after prolonged RA treatment show significant overlap with genes down-regulated after LSD1 inhibition. When looking at proximity of genes down-regulated after prolonged RA treatment to LSD1-bound enhancers in the different groups (G1, G2, G3), we found no significant overlap. Two conclusions can be drawn from this analysis: First, the analysis further strengthens the conclusion that LSD1 inhibition and prolonged RA treatment affect the expression of similar genes. Second, the observation that genes down-regulated by RA are not in proximity to LSD1-bound enhancers underscores our conclusion that LSD1 exerts direct effects only on the up-regulated genes. We included the analysis as **Suppl. Fig. 6f** in the revised submission.

6. RA late treatment seems to give rise to multihormonal cells (fig 5b)? This deserves comment. The authors could try to treat pancreatic explants with RA early and late and look at endocrine cells to see both whether they can recapitulate LSD1 knockout data but also whether RA treatment at later stages results in polyhormonal cells. This could be really important for understanding the mechanisms that segregate endocrine subtypes and might be relevant for the generation of mono-hormonal insulin-producing beta cells for diabetes.

Response:

One of the limitations of the hESC differentiation system is that a disproportionate percentage of endocrine cells is polyhormonal even without RA treatment. It is simply a feature of the hESC differentiation system. Whether these polyhormonal cells are representative of polyhormonal cells that are transiently observed during early mouse pancreas development is unclear. For this reason, we have been hesitant to compare this aspect of the phenotype in the hESC system and in mouse embryos.

We conducted an analogous experiment to the hESC “late RA treatment” in mouse explants, by explanting pancreatic buds at e12.5 and culturing the explants in a high concentration (1 μ M) of RA. This is the same concentration of RA as used for pancreatic fate induction (PMID: 25211370) and “late RA treatment” in our manuscript. The presence of 1 μ M RA had no obvious effect on endocrine cell numbers or the proportion of polyhormonal cells (**Reviewer Fig. 2**), suggesting that the effects of late RA treatment of mouse explants mirror the effects of late RA treatment in the hESC system. To also test the effects of “early RA treatment” on mouse explants, we treated e10.5 pancreatic explants with 1 μ M of RA, which is the analogous concentration to the hESC experiment. In our hands, e9.5 or e10.5 pancreatic explants were not viable under these conditions which precluded assessment of effects with the dosage we used in hESCs.

However, there is evidence in the literature that RA treatment of early mouse pancreatic explants inhibits endocrine cell differentiation (PMID: 18665267). In this study by Öström et al., mouse pancreatic explants from e10.5 embryos were treated with a low (25 nM) or high dosage (50-100 nM RA) of RA. While low dose (25 nM) RA treatment improved endocrine cell differentiation, higher dose RA treatment inhibited endocrine cell differentiation. Thus, higher dosages appear to inhibit endocrine cell differentiation similar to what we observed in hESC-PP1 cells. Given the supportive evidence from the study by Öström et al. (PMID: 18665267), we added a sentence to the discussion citing their finding that RA at higher concentrations inhibited endocrine cell differentiation in early pancreatic bud explants (line 382): “*Similarly, exposure of mouse embryonic pancreatic explants to high RA concentrations prior to endocrine cell differentiation has been shown to impair endocrine cell formation (PMID: 18665267).*” With regard to effects of RA on polyhormonal cells, for reasons stated above, we would prefer to not

include the polyhormonal cell comparison between mouse embryos and hESC-derived cells in the final manuscript.

Reviewer Figure 2. Effects of “late” RA treatment on mouse pancreatic explants. Pancreatic buds were explanted at e12.5 and treated for 24 hours with 1 μ M RA before analysis. Immunofluorescent staining for insulin (Ins) and glucagon (Gcg).

7. Fig 6f: why do you have so many LSD1 positive cells in LSD1Delta-early (around but also far from insulin+ cells)? Is this picture representative?

Response:

We apologize that the images did not clearly convey the abundance of Lsd1 protein in the pancreas of *Lsd1 Δ early* embryos. The bottom row images in **Fig. 6f** are high magnification images where we specifically focused on areas of hormone⁺ cells to determine whether the presence of hormone⁺ cells in *Lsd1 Δ early* embryos can be attributed to mosaic deletion with the tamoxifen-inducible system. These high magnification images serve to illustrate that the vast majority of hormone⁺ cells in *Lsd1 Δ early* embryos (quantified in **Suppl. Fig. 8e**) escaped Cre recombination and therefore express Lsd1. The lower magnification images included below (**Reviewer Fig. 3**) show reduced but not absent staining of Lsd1 throughout the pancreas of *Lsd1 Δ early* embryos. This demonstrates that tamoxifen induced recombination mosaically. In the panel below, we also show yellow fluorescence signal (YFP) from the *Rosa26-stop/flox-eYFP* allele which we included in the mouse cross to trace cells that underwent Cre-mediated recombination. We found YFP⁺ cells in the pancreas to be mostly Lsd1-negative, demonstrating that a large proportion of pancreatic epithelial cells underwent recombination. Upon the reviewer’s request, we will be happy to include the low magnification images and YFP tracing into the manuscript.

Reviewer Figure 3. Mosaic recombination in *Lsd1 Δ early* embryonic pancreas. Immunofluorescent staining for Lsd1, chromogranin A (Chga) and YFP at e18.5 in control and *Lsd1 Δ early* mice. Scale bar, 50 μ m.

Minor points

1. It would be helpful to add in figure 1 the days to reach the different intermediates

Response:

As suggested, we marked the days to each differentiation step in **Fig. 1a**.

2. Is LSD1 ubiquitous? What about other tissues in which RA plays an important and biphasic role? For ex in the nervous system, RA is first involved in posteriorization and then it is essential for neuronal differentiation. The authors should add something about this in the discussion.

Response:

We thank the reviewer for the suggestion. LSD1 is indeed widely expressed in development and adult tissues. We added a paragraph to the discussion, highlighting possible parallels of LSD1's role in pancreas and neuronal development (line 390).

“Similar to pancreas development, LSD1 is expressed in the developing nervous system and plays an important role in neural differentiation (PMID: 20123967, 25018020). Given that RA has time-dependent roles during different phases of neural development (PMID: 17882253), it is possible that a similar connection between LSD1 and the regulation of RA signalling windows exists during neurogenesis.”

3. If RXR ChIPseq has been undertaken and is referred to, the data about location of peaks etc. should be made available.

Response:

We apologize for the oversight and added the RXR peaks as **Suppl. Table 1c**. We also added the data set to the GEO submission.

Reviewer #2

In this manuscript the authors, using a well-defined stepwise differentiation program of hECS into pancreatic endocrine cells along with relevant genomic approaches, suggest that LSD1 affects the epigenetics landscape of the differentiation program in a defined window and demonstrate that LSD1-mediated enhancer decommissioning affects retinoic acid (RA)-induced expression. The authors also show that, in the absence of LSD1, RA-dependent enhancers remain active despite the absence of RA, with a consequent failure to progress into the differentiation program. Collectively, their findings indicate that LSD1 functions at a defined window of the differentiation program, and loss of LSD1 alters epigenetics landscape affecting RA signaling.

The findings reported are strongly indicative of the role of LSD1 in the appropriate control of endocrine program and highlight a novel time-dependent function of LSD1 in endocrine differentiation. I share with the authors the possibility that their findings might be broadly relevant.

Overall, the authors have made potentially interesting findings but, some data presented in the MS are over-interpreted and there are several areas where it falls somewhat short. Some additional experiments as well as a more accurate presentation of the data will definitively improve the significance of their observations.

Response:

We appreciate that the reviewer perceives our findings on the role of LSD1 in endocrine cell differentiation as important and broadly relevant. We are also grateful for the constructive suggestions for how to further improve this work. As suggested by the reviewer, we validated findings from the LSD1 inhibitor experiment by *LSD1* knockdown in hESCs, excluded a drop in FAD levels as an explanation for the temporal window of LSD1 function, and identified transcription factors that co-occupy LSD1-regulated enhancers and likely recruit LSD1 to these enhancers.

Specific comments

1. A compounding issue is that the authors consider the use of drugs such as the irreversible TCP and the reversible SP2509 (along with GSK-LSD1 and GSK2879552) equivalent to LSD1 knock-out and this is not fully correct. These drugs, especially TCP (used to generate most of the data), also inhibit the activity of other proteins. This point raises the question of the selectivity of the LSD1 inhibitors since TCP in addition to inhibit monoamine oxidases MAOA/B, also affects the LSD1 paralog LSD2. I appreciate the *in vivo* data presented in Figure 6 supporting the specific role of LSD1 in endocrine cell formation, however, they are insufficient to validate all data obtained in the *in vitro* studies using pharmacological inhibition of LSD1. It is pertinent to consider that like LSD1, LSD2/KDM1B is a FAD-dependent enzyme histone demethylase and it participates to gene transcription at least in part through control of transcription elongation control. I would urge the authors to determine the specificity of LSD1 inhibition at PP1-PP2 stage by CRISPR-Cas9 editing (or similar specific LSD1 genetic ablation). Alternatively, if LSD2 is expressed in hESCs, (likely to be the case), LSD2 ChIP-seqs in the presence and absence of TCP would help to segregate the relative contribution of both KDM1A and KDM1B at PP1 to PP2 transition.

Response:

We agree that pharmacological inhibition of LSD1 is not equivalent to a genetic knockout or knockdown. To validate our findings obtained in the hESC system with inhibitors of LSD1, we transduced PP1 stage cells with lentiviruses expressing two different *LSD1* shRNAs or a

scramble control shRNA. Both shRNAs led to a significant decrease in *LSD1* mRNA expression when compared to a scramble control (**Suppl. Fig. 3c**). Analysis of endocrine stage cultures by FACS for insulin as well as qRT-PCR and immunostaining for insulin, glucagon, and somatostatin revealed a drastic reduction in hormone gene expression and the number of hormone⁺ cells after *LSD1* shRNA transduction (**Suppl. Fig. 3a-c**). These results confirm that the phenotype is *LSD1*-dependent and strengthen the conclusion that the hESC in vitro model and *LSD1* inhibitors accurately mirror *LSD1*'s in vivo role in endocrine cell differentiation. The experiment was repeated on independent differentiations three times with consistent results.

2. ChIP-seqs for *LSD1* were carried out at PP1 stage with or without *LSD1* inhibitor (I assume TCP??). For how long the cells were treated with TCP before chromatin isolation? Specifically, I wonder whether the chromatins from treated and untreated cells were prepared at the same time of differentiation protocol. As it stands the *LSD1* inhibitor protocol is unclear.

Response:

We apologize for not clearly explaining how the experiments were performed. ChIP-seq experiments for *LSD1* were performed at the PP1 and PP2 stages without TCP treatment. We added a clarification on when ChIP-seq experiments were conducted to the methods section (line 531):

“ChIP-seq experiments for histone modifications were performed on day 10 with no TCP treatment or after TCP treatment (treatment on days 7, 8, and 9). ChIP-seq experiments for LSD1 were conducted on day 5 (GT stage) day 7 (PP1 stage) and day 10 (PP2 stage) without addition of TCP.”

3. Previous work in mouse ESCs (Whyte et al. 2012) reported *LSD1* signals at core promoter regions. In addition, several reports indicate the presence of *LSD1* signals at TSS. From the presented data it is not clear how many of the 15,084 *LSD1* peaks are at TSS? And..... if there are examples where *LSD1* is present in both distal and core TSS, or only at distal, or only at TSS. Please provide these informations.

Response:

Out of the 15,084 *LSD1* peaks 3,285 are proximal (within 3kb of TSS). Thus, the vast majority of peaks is at distal enhancers. We included this information as **Suppl. Fig. 4a** and **Suppl. Table 1a,b** (a, proximal peaks; b, distal peaks). In our study, we defined genes as associated with an enhancer if the enhancer is localized within ± 100 kb from the TSS. When applying these criteria, there are genes that have an *LSD1* peak at the TSS and at associated enhancers. However, within the set of genes that were RA-induced and up-regulated after *LSD1* inhibitor treatment (e.g. *HOXA1 HOXA3, HOXC4, CYP26A1, CYP26B1*, and other genes shown in **Fig. 4h**), no *LSD1* peaks were detected at the TSS. Therefore, up-regulation of this group of genes after *LSD1* inhibitor treatment cannot be explained by regulation due to *LSD1* binding at promoters and therefore is likely due to effects of *LSD1* on distal enhancers. To convey this point, we added the following sentence to the discussion section (line 406):

“It is possible that LSD1 at promoters also contributes to gene regulation after LSD1 inhibition. However, TSSs of the majority of RA-induced genes that were up-regulated after LSD1 inhibition (e.g. HOXA1 HOXA3, HOXC4, CYP26A1, CYP26B1) were not LSD1-bound (Supplementary Table 1a), suggesting that effects on gene expression are mediated by distal enhancers.”

4. The authors state that they:..... isolated enhancers that are active at PP1 and/or PP2 and also bound by *LSD1* at PP1,These putative enhancers were divided in 3 groups G1, G2 and G3, on the basis of the relative abundance of H3K27ac. I assume that the authors imply that the

distal LSD1-bound regions (enhancers) are also co-occupied by hESC master transcription factors such as Oct4, Sox2, Myc and Nanog and the Mediator coactivator. I reckon that this assumption must be validated. It will be sufficient to demonstrate by ChIP data that a specific distal LSD1 binding region (such as HOXA1 shown in Fig 2 E) is also co-occupied by a dedicated TF or Mediator factor.

Response:

We thank the reviewer for the constructive suggestion. To characterize the complex in which LSD1 resides at enhancers, we conducted ChIP-seq experiments to determine whether early pancreatic transcription factors, such as FOXA1, FOXA2, GATA4, GATA6, and HNF6, occupy the same enhancers as LSD1 in PP1 stage pancreatic progenitors. We found that binding of these pancreatic transcription factors was enriched at LSD1-bound enhancers (**Suppl. Fig. 4b**). We can conclude from this data that early pancreatic lineage-determining transcription factors likely recruit LSD1 to pancreatic enhancers. We note that Oct4, Sox2, Myc, and Nanog are not expressed in pancreatic progenitors, which precludes an analysis of their co-binding with LSD1 at the pancreatic progenitor cell stage. A previous study by Young and colleagues (PMID: 22297846) examined LSD1 binding in undifferentiated human ESCs and found that LSD1 localizes to similar sites as Oct4, Sox2, and Nanog in human ESCs. Therefore, LSD1 recruitment is context-specific and depends on the specific repertoire of expressed lineage-determining transcription factors.

In the revised manuscript, we added the data showing enrichment of FOXA1, FOXA2, GATA, GATA6, and HNF6 binding at distal LSD1 sites as **Suppl. Fig. 4b**. We also added genome browser snapshots for the *HOXA1*, *CYP26A1*, and *CYP26B1* loci in **Suppl. Fig. 4c**.

5. LSD1 is a FAD-dependent enzyme. Is the FAD level constant throughout the hESC differentiation program toward pancreatic endocrine cell lineage? The rate-limiting role for FAD in LSD1-mediated effects has been recently documented by Yang et al. EMBO J. 2017, 36: 1011–1028, doi:10.15252/embj.201694408. Reduction of FAD levels at PP2 stage will provide a plausible explanation of the temporal window of LSD1 function.

Response:

We thank the reviewer for this insightful comment. As suggested, we measured FAD levels throughout the hESC differentiation time course. FAD levels were similar at each stage of differentiation, with FAD levels trending up at the PP2 stage. Therefore, reduced FAD levels are unlikely to account for the temporal requirement of LSD1 in endocrine cell differentiation. We added these data as **Suppl. Fig. 1c**.

6. In fig 1 the effects of LSD1 (both early and late) at EN stage is revealed only by flow cytometry. I suggest that the authors validate the phenotype also by qRT-PCR, as shown in panel A of Supplementary Fig 5.

Response:

The results of the qRT-PCR are shown in **Fig. 1c**.

7. Provide qRT-PCR of LSD1 or WB in Supp. 1a (no FPKM). It is not clear if the LSD1 wave of expression (low in DE, high in GT) is significant. Similarly, data from Fig. 4 panel H, and Supplementary Fig 6 E, should be validated by qRT-PCR.

Response:

As suggested, we confirmed the results from our RNA-seq analysis by qRT-PCR (**Reviewer Fig. 4**). Consistent with the RNA-seq results, *LSD1* mRNA levels increased slightly from DE to GT and then remained stable from the GT to the EN stage (**Reviewer Fig. 4a**). It is possible that

the increase in *LSD1* mRNA levels during the DE to GT transition has biological significance. However, this is not of relevance to our study, because we examined the role of *LSD1* during the PP1 to PP2 and PP2 to EN transitions. During this time window, *LSD1* mRNA levels are stable.

We further confirmed key results from Fig. 4h and Suppl. Fig. 6e (now **Suppl. Fig. 7f**) by qRT-PCR (**Reviewer Fig. 4b,c**). As observed in RNA-seq, known RA target genes, such as *HOXA1*, *HOXA3*, *HOXC4*, *CYP26A1*, and *CYP26B1*, were induced by both prolonged RA treatment and *LSD1* inhibition. Likewise, *CLIC6*, *IGFBP6*, *LOC400043*, and *SPOCK2* showed the same pattern as observed by RNA-seq. The only two exceptions were *OSR1*, which did not show significant induction by *LSD1* inhibition in the qRT-PCR assay and *PRLHR*, which showed no significant increase after prolonged RA treatment in qRT-PCR. We conclude that for most genes the qRT-PCR analysis confirms the RNA-seq results. We obtained similar confirmatory results for the genes shown in **Suppl. Fig. 7f (Reviewer Fig. 4c)**.

We would prefer to leave the RNA-seq results in the manuscript. Since showing both results will require additional supplementary figures, we have not included the qRT-PCR data into the revised manuscript but will add the data upon the reviewer's request. For the measurement of *LSD1* mRNA levels during the differentiation time course, we added additional RNA-seq data and now show results from three replicates in **Suppl. Fig. 1b**.

Reviewer Figure 4. qRT-PCR confirmation of RNA-seq results. (a) qRT-PCR for *LSD1* at definitive endoderm (DE), gut tube (GT), early pancreatic progenitor (PP1), late pancreatic progenitor (PP2), and endocrine cell (EN) stages. **(b,c)** qRT-PCR for indicated genes and conditions from Fig. 4h (b) and Suppl. Fig. 7e (c). n = 3.

8. Legend Supp. Fig. 3. CYP26A1 is duplicated.

Response:

We thank the reviewer for noticing the mistake, which we have corrected.

9. To provide sufficient details for the work to be reproduced successfully, the authors should add details of the pharmacological conditions used.

Response:

We carefully revised the Methods section and also refer to the reporting summary which lists all reagents and their validation. In addition, we amended the Figure Legends to indicate which specific reagents were used.

10. The description of figures is often rather cryptic. The figure legends need to be specified so that one can understand what was done. As it stands too much guessing is necessary.

Response:

We expanded the Figure Legends to better explain the experiments and also indicated for each panel how many replicates were analyzed.

Reviewer #3 (Remarks to the Author):

This manuscript provides evidence that the histone demethylase LSD1 regulates the epigenetic status of developmental enhancers to control how those enhancers respond to extracellular signals, namely retinoic acid during pancreas specification. While LSD1 is known to “decommission” enhancers in the context of cell signaling the current study suggests that this provides a mechanistic basis for developmental competence and how the duration of transcriptional responses to extracellular signals are modulated and reset during lineage progression. This is an important area of work that in principle the findings represents an important advance in the field. However, there are a few important issues that need to be addressed to support the model.

Response:

We appreciate that the reviewer recognizes the novelty of our finding that LSD1 modulates the duration of transcriptional responses to extrinsic signals. As suggested by this reviewer, we expanded our analysis of LSD1 binding to the gut tube stage and identified transcription factors that recruit LSD1 to pancreatic enhancers. Furthermore, we strengthened the link between LSD1 and regulation of RA signaling by showing that *HoxA1* is upregulated after *Lsd1* deletion in mouse pancreatic buds.

1. What recruits LSD1 to enhancers and when is it first there. If the model is correct then LSD1 should not be at G1 enhancers at the gut tube stage. If it is at there, then it should prevent those enhancers from becoming active in response to RA signaling. The authors really need to show LSD1 binding at the GT-stage to address this point as is it a key aspect of the model.

Response:

We thank the reviewer for the insightful comment. LSD1 is expressed throughout the entire time course of hESC differentiation (see **Suppl. Fig. 1a,b**) and our data suggest that its recruitment to enhancers is context-dependent. We addressed the question of what recruits LSD1 to early pancreatic enhancers by conducting ChIP-seq experiments to determine whether binding of early pancreatic transcription factors, namely FOXA1, FOXA2, GATA4, GATA6, and HNF6, is enriched at LSD1-bound enhancers in PP1 stage pancreatic progenitors. We found that LSD1 binding sites indeed coincide with the binding of these pancreatic transcription factors, suggesting that early pancreatic lineage-determining transcription factors recruit LSD1 to pancreatic enhancers. In the revised manuscript, we added data showing enrichment of FOXA1, FOXA2, GATA4, GATA6, and HNF6 binding at distal LSD1 sites as **Suppl. Fig. 4b**. We also added genome browser snap shots for the *HOXA1*, *CYP26A1* and *CYP26B1* loci in **Suppl. Fig. 4c**.

As suggested by this reviewer, we further analyzed LSD1 binding at each class of pancreatic enhancers at the GT stage and determined whether LSD1 occupies these enhancers prior to pancreas induction. Consistent with our model, LSD1 occupies G1 enhancers to a much lesser extent at the GT stage than at the PP1 stage (**Fig. 2c,e** and **Suppl. Fig. 4e**). The lack of LSD1 occupancy at the GT stage was clearly evident at RA-responsive *HOXA1*, *CYP26A1* and *CYP26B1* enhancers (**Fig. 2e** and **Suppl. Fig. 4e**). Importantly, in contrast to G1 enhancers, LSD1 occupied enhancers of the G2 and G3 class at the GT stage (**Fig. 2c**). Thus, the analysis of LSD1 binding at the GT stage further strengthens our model and conclusions.

2. The idea that LSD1 regulates competence of an enhancer to respond to RA-RxR activity is attractive, but the data that the authors show is mostly correlative. If the model is correct then ectopic LSD1 should prevent (or attenuate) RA from activating these enhancers.

Response:

We appreciate the reviewer's suggestion. We have thought extensively about experiments that would strengthen the connection between LSD1 and the activity of RA-responsive enhancers. Unfortunately, the proposed LSD1 gain-of-function experiment will not be able to prove or disprove our model. The reason is as follows: In the presence of RA, the RA receptor recruits HATs to the transcriptional complex and the activity of HATs leads to histone acetylation, a mechanism that has been extensively documented for the RA receptor (reviewed in PMID: 25560970). Therefore, when we add RA to the culture medium to activate pancreatic RA-responsive enhancers after the GT stage, HATs are recruited to these enhancers. It has been shown that histone acetylation overrides LSD1 activity and that in the presence of HATs and acetylated histones, LSD1 does not demethylate its substrates (PMID: 22297846). Mechanistically, LSD1 itself has been shown to be the substrate of HATs and HDACs. LSD1 acetylation dampens LSD1 recruitment to nucleosomes and/or LSD1 stability (PMID: 27292636; 27212032; 27977115). Thus, LSD1 can only decommission enhancers when HATs are no longer recruited (in our system this is after RA removal). Since LSD1 will not attenuate enhancer activity in presence of RA, addition of ectopic LSD1 during enhancer activation by RA will not be able to prove or disprove our model. Furthermore, LSD1 is expressed throughout the entire differentiation time course (**Suppl. Fig. 1a,b**), and it is difficult to predict how increasing LSD1 protein levels would affect its recruitment to specific sites.

Given these limitations, we reasoned that the best way to test the question of regulation of competence would be by re-exposing cells to RA after RA has been withdrawn (see **Fig. 5a**) and analyzing the expression of RA-responsive genes. The analysis showed that prior LSD1 inhibitor treatment augmented the level of induction of RA-responsive genes to some extent (**Fig. 5d**). We agree that the evidence is somewhat indirect but think that this was the best possible experiment to test the question of regulation of competence. Overall, the regulation of competence (i.e. responsiveness of the enhancer after exposure to RA) is a minor aspect of our study which we do not mention in the abstract. To point out where evidence is correlative and where conclusive, we have edited the manuscript substantially and have pointed out limitations.

3. The data and interpretation in Fig. 2 d-f is confusing. The model proposes that LSD1 demethylates after the enhancer after RA is withdrawn from the culture and that in the LSD1i demethylation doesn't happen and RA-responsive genes expression persists. But if this is true then I would expect an increase in K27ac in the LSD1i to accompany the persistent gene expression. How do you get persistent expression in the absence of K27Ac?

4. The evidence that the modest ~2-fold increase in me2 on G1 peaks can account for the persistent expression is not very strong. These two observations might be true-true and unrelated. An alternative explanation is that LSD1 is doing something else, independent of K4me2 levels to regulate expression.

Response to #3 and #4:

Clearly, H3K27ac is the histone modification that has been most closely associated with enhancer activity and transcription of associated target genes. Whether in some contexts H3K4me2 at enhancers is sufficient to drive target gene transcription is less clear. There is evidence that H3K4me2 deposition is dependent on transcription at enhancers, and it has been shown that these enhancer RNA transcripts correlate with the expression of nearby genes (PMID: 23932714). These findings suggest a correlation between H3K4me2 at enhancers and gene transcription, which is consistent with our observations. However, G1 enhancers also show some H3K4me2 deposition at the GT stage (**Fig. 2d**) without significant effect on gene transcription. One possibility is that there are differences in 3D chromatin architecture at these enhancers between the GT and PP2 stage and that these differences determine the impact of

H3K4me2 on gene transcription. We feel, however, that such analysis is beyond the scope of this study.

While we cannot determine the exact mechanism that underlies gene activation in the presence of di-methylated enhancers, we conducted additional analyses to more tightly link the persistence in expression of RA-responsive genes after LSD1i treatment to changes in chromatin. To this end, we analyzed H3K4me3 signal at TSSs (± 2.5 kb) of genes up- or downregulated after LSD1 inhibitor treatment. As shown in **Reviewer Fig. 5**, LSD1i treatment was accompanied by an increase in H3K4me3 signal at the TSSs of upregulated genes. By contrast, no difference in H3K4me3 signal was observed at TSSs of genes downregulated after LSD1i treatment. This shows that increased H3K4me2 deposition at enhancers correlates with increased H3K4me3 deposition at promoters of genes that are upregulated after LSD1i treatment. H3K4me3 deposition at promoters is closely associated with gene transcription. Thus, LSD1i treatment has effects on chromatin that are consistent with active gene transcription. Future experiments will have to determine the exact mechanism by which these histone modifications regulate gene expression. We added the following statement to the discussion (line 422):

“Our observations are consistent with evidence that H3K4me2 deposition is dependent on transcription at enhancers, and that enhancer RNA transcripts correlate with the expression of nearby genes (PMID: 23932714). The exact contexts in which H3K4me2/me1 can drive gene expression and the mechanisms employed will require further studies.”

Because the analysis of H3K4me3 at promoters does not directly address how H3K4me2 at enhancers leads to gene activation, we did not include the H3K4me3 data in the revised submission. However, we will be happy to do so upon the reviewer’s request.

Reviewer Figure 5. H3K4me3 signal at TSSs of genes up- and downregulated after LSD1i treatment. Tag density plots of H3K4me3 tag distribution at TSSs of gene upregulated (left) and downregulated (right) with and without early LSD1 inhibition (LSD1i^{early}) in hESC cultures.

5. In Fig. 4h it would be good to also show the expression of these key LSD1/RA-responsive genes at PP1 so that the reader has a clear understanding of what this elevated expression at PP2 is relative to that see normally at the PP1 stage. The elevation could be very modest only 10% of PP1 levels. Fig. 3d is a good stat but the authors should show the expression of the key LSD1/RA-targets. Similarly in Fig. 5 it would be good to show the RNA-seq/RT-PCR data in a histogram to more clearly show the expression of key LSD1/RA-responsive genes at PP1, PP2 and EN +/-late RA to be able to compare the level of expression. Are the LSD1/RA-regulated genes really not activated at all when RA is added late (it is hard to extract that info from the

current figure). When they are activated in the LSD1i+RAlate condition what is the level of expression relative to that normally seen at PP1.

Response:

As suggested, we included expression data from the PP1 stage into **Fig. 4h**.

We apologize if findings from **Fig. 5d** were not clearly conveyed. Our data do not show that late stimulation with RA has no effect on LSD1/RA-regulated genes. In **Fig. 5d**, the Y-axis shows the log₂ fold change in expression as a result of late exposure of the cells to RA. “0” indicates no change in response to RA and a value of 1.0 on the Y-axis indicates a 2-fold change in response to RA (a 0.5 log₂-fold change would be a 1.4-fold change). The plot shows that most genes exhibit an up to ~1.5-fold induction with late RA exposure. On the X-axis, we plotted the log₂-fold change in expression after late RA treatment of cells previously treated with the LSD1 inhibitor. Any gene to the right of the dotted black line has a greater fold-change in response to late RA exposure with prior LSD1 inhibitor treatment compared to late RA treatment alone. We had to plot data as a fold-change rather than absolute values because each of the late RA treatment conditions has a different control owing to the fact that early LSD1 inhibitor treatment eliminates the endocrine cell population and this introduces a population bias independent of LSD1 inhibitor effects. As suggested by the reviewer, we plotted FPKM values of the RA-responsive genes (exemplified by HOX transcription factors) in PP1 cells compared to levels in PP2 cells and EN cells treated with RA late ± prior LSD1i treatment. These data are shown in **Suppl. Fig. 7e**. For some genes (i.e. *HOXA3* and *HOXC4*) levels after late RA exposure + prior LSD1i treatment are close to levels at PP1; for others, levels are higher in PP1 (i.e. *HOXA1*). It is important to note that the effect of prior LSD1i treatment is modulatory rather than an “on-off switch” as evident from **Fig. 5d** and **Suppl. Fig. 7e**. We revised the text to state this more clearly:

Line 304: “Thus, *LSD1* appears to dampen, although not obliterate, future RA responsiveness in cells that have been previously exposed to RA.” Line 308: “We note that other factors must also control RA responsiveness since the effect of prior *LSD1* inhibition is small.”

Overall, the modulatory effect of LSD1 inhibition on RA responsiveness is a minor aspect of our study which we also do not mention in the abstract.

6. If the LSD1i or RA extended cells they are stuck at a progenitor state then PCA analysis of the RNA-seq might show that the PP2-LSD1i or PP2-extended RA are more similar to PP1 cells.

Response:

We conducted PCA analysis, comparing PP1 cells, PP2 cells, PP2 cells treated with LSD1i, and PP2 cells with extended RA treatment. The analysis revealed that the extended RA treatment had overall the smallest effect on the transcriptome of PP2 cells. In comparison, LSD1i-treated PP2 cells exhibited greater differences to untreated PP2 cells. This is consistent with 1400 genes being differentially expressed in PP2-LSD1i cells compared to only 165 differentially expressed genes in RA-extended PP2 cells. PP1 cells were distinct from PP2-LSD1i and PP2-extended RA cells, showing that there is no general developmental arrest of PP1 cells after these treatments. We included these data as **Suppl. Fig. 6e** in the revised manuscript.

7. For the mouse genetics any evidence of elevated RA-target gene (*Hox*) expression in the LSD1 KO?

Response:

We agree that demonstration of increased *Hox* gene expression would provide important evidence that the in vitro hESC model accurately reflects the role of *Lsd1* in vivo. To this end,

we conducted RNAscope in situ hybridization for *HoxA1* mRNA on pancreatic sections from *Lsd1*^{Δearly} embryos. The analysis revealed a clear increase in *HoxA1* signal in the pancreatic epithelium of *Lsd1*^{Δearly} compared to control embryos. Thus, as in the hESC system, *Lsd1* deletion in early pancreatic progenitors in mice leads to up-regulation of RA-responsive genes of the *Hox* transcription factor family. We included these data as **Fig. 6g** in the revised submission (also shown below).

Minor points

1. Need to report the antibodies used for ChIP

Response:

We apologize for not including this information into the first submission. All antibodies used in our study are listed in the “Reporting Summary” which were submitted to *Nature Communications* as a separate document.

2. While the enrichment analysis of motifs is okay, the authors should show the motif analysis of LSD1 ChIP peaks from G2, G3 and the rest of the genome other than G1-3. Perhaps RxR is just a common feature of all of these.

Response:

We conducted the proposed analysis of motifs enriched at enhancers belonging to G2 and G3 using the genome minus G1-3 enhancer regions as background. The RXR motif was not enriched in this comparison, confirming our conclusion that the RAR-RXR motif is specifically enriched at G1 enhancers. We included this analysis as **Suppl. Table 6b** and added the following statement to the results section (line 240):

“When motifs at G2 and G3 enhancers were compared against the entire genome, excluding G1, G2, and G3 regions, no RAR/RXR motif enrichment was observed, further supporting specific enrichment of the RAR/RXR motif at G1 enhancers (Supplementary Table 6b).”

3. The authors definition of decommissioned and deactivated enhancers changes a bit at different points in the paper. G1 enhancers are selected as “decommissioned” based on loss of K27ac. But then they classify decommissioned based on the removal of me2, in the context of no change in K27ac.

Response:

We appreciate the comment and have revised the manuscript to ensure that the terminology is used consistently.

Reviewer #4

This is a clearly written paper with some interesting and novel findings. Generally, the experiments are well performed and many of the conclusions drawn are justified. However, the main conclusion is not supported by the data in their present form. Thus, I would like to see the following major issues addressed before recommending publication:

Response:

We appreciate that the reviewer recognizes the novelty of our finding that LSD1 modulates the duration of transcriptional responses to extrinsic signals. As detailed below, we have taken substantial effort to address this reviewer's comments.

1) The motif search done by the authors, which revealed the RA-response element being enriched is not particularly impressive given that Table S6 shows that only 1.14% of the LSD1 target sequences had this motif. Adding the DR1 motif (line 20 in Table S6) at 2.4% only slightly improves this. It is thus surprising that nearly one third of the G1 enhancers showed RXR binding, but without access to the ChIP-seq data, one cannot assess the significance of this observation. The GEO data set should therefore be made available to the reviewers.

Response:

We agree that the percentage of LSD1 target sequences matching the motif is much lower than the percentage of sites with RXR binding. One possible explanation is that RXR also binds to sites with imperfect motif match, which is true for many TFs. We observed RXR binding at G1 enhancers near *HOXA1*, *CYP26A1* and *CYP26B1*, and have added genome browser tracks in **Suppl. Fig. 4c**. We apologize for not including the RXR data set into the first GEO submission. We added the data set to GEO and also included the list of distal RXR binding peaks as **Suppl. Table 1c**.

2) The phenotypic similarity between LSD1 inhibition and prolonged RA exposure is, albeit interesting, mostly correlative in nature and none of the experiments directly test whether the effect of LSD1 inhibition is solely caused by a failure to decommission RA-responsive enhancers and thus prolonged activity of the genes regulated by those enhancers. Thus, the link between LSD1 and RA signaling is somewhat tenuous. To bolster their conclusions the authors should test whether inhibition of RA signaling, for example with an inverse agonist that prevents activation even from "open" enhancers, can rescue the effect of LSD1 inhibition. This should be done both in the hESC cultures and in explants from Lsd1-deficient embryonic mouse pancreas.

Response:

We agree that a rescue experiment would be ideal to prove that misexpression of RA-induced genes causes the LSD1 loss-of-function phenotype. We extensively discussed a possible experiment along the lines of what this reviewer proposes. Unfortunately, inhibiting RA signaling cannot answer the question, which we hope the schematic below helps illustrate. An inverse agonist to RA would reverse the effects of RA while RA is present in the culture medium, but would have no effect after RA is removed from the medium. The addition of RA (through recruitment of HATs) acetylates RA-dependent early pancreatic enhancers. Once RA is removed (after Day 8 (D8) of differentiation), RA-dependent enhancers are deacetylated and this deacetylation event enables LSD1 to demethylate histones. Thus, LSD1 acts on its histone substrates only after RA is removed. Since histones are already deacetylated when the LSD1 inhibitor acts to prevent histone demethylation, addition of an RA inverse agonist (depicted as antagonist in schematic), will have no effect on LSD1-mediated histone demethylation which occurs from Day 8 to Day 10. Therefore, the RA inverse agonist would not prevent persistence

of H3K4me2 marks on enhancers in the presence of the LSD1 inhibitor and the increase in H3K4me2 caused by LSD1 inhibition will not be reversed. We tried the experiment anyway and added an inverse RA agonist to the medium from Day 7 to Day 10. As expected, endocrine cell differentiation was not rescued. Upon the reviewer's or editor's request, we could include the results in the manuscript. However, given that a rescue is not expected, we feel that the result does not add much to the study.

For the revision, we have tried to strengthen the link between LSD1, the regulation of RA signaling, and endocrine cell differentiation by obtaining other evidence. First, we provide additional evidence for the similarity in phenotypes by showing that NGN3 expression is unaffected by both LSD1 inhibition and prolonged RA treatment in the hESC system and in mice (see below). Second, we extended the analysis of LSD1 and transcription factor binding to show that pancreatic transcription factors recruit LSD1 to enhancers of RA-responsive genes at the transition from gut tube to pancreatic progenitors, whereas LSD1 occupies other classes of enhancers already at the gut tube stage (**Fig. 2c,e** and **Suppl. Fig. 4b,c,e**). This further links the time window of LSD1 inhibitor effects to the regulation of G1 enhancers and RA-dependent genes. Third, we provide evidence for upregulation of RA-responsive genes after *Lsd1* deletion in mice by showing that the RA target gene *HoxA1* is upregulated (**Fig. 6g**). We feel that despite the lack of direct evidence for rescue of the phenotype, our study still provides ample novelty and mechanistic insight. Our study is the first to identify LSD1 as a time window-dependent regulator of endocrine cell differentiation, to show that this role is conserved from mice to man, and to demonstrate that LSD1 is required for epigenetic silencing of RA-dependent enhancers to limit the duration of signal-dependent gene activation during developmental progression. Our identification of a cell-intrinsic epigenetic feedback mechanism by which the duration of a response to a developmental signal is limited is novel and likely of broad relevance across tissues. To not overstate our conclusions, we have revised the manuscript and changed the title and abstract.

3) Endocrine cells differentiate from pancreatic progenitor cells via a NEUROG3 expressing precursor stage, yet nothing is mentioned about this in the paper. In the discussion the authors refer to “the endocrine-specified pancreatic progenitor cell stage” but it is unclear what they mean by this. Do the authors mean NEUROG3+ cells here? More importantly, the authors make no attempt to decipher whether the effect of LSD1 inhibition

or extended RA treatment on EN cell differentiation is executed by preventing induction of NGN3 or preventing the differentiation of NGN3+ cells into EN cells. The tools for making this distinction are readily available and have previously been used by the authors. This is of major importance for the field and for the usefulness of these findings in the translation to hESC differentiation protocols. Thus, this omission should be remedied.

Response:

We appreciate the comment and conducted NGN3 staining in hESC-derived pancreatic progenitors treated with the LSD1 inhibitor, in hESC-derived pancreatic progenitors exposed to extended RA, and in *Lsd1*^{Δpan} mouse embryos. In all three models, NGN3 expression was maintained. This demonstrates that LSD1 inhibition or loss specifically affects the differentiation of endocrine progenitors but does not affect endocrine lineage commitment. Furthermore, the data underscore the similarity in phenotype between LSD1 inhibition and prolonged RA exposure, which strengthens the overall conclusions of our manuscript. We included NGN3 staining and *NGN3* mRNA expression analysis as **Suppl. Fig. 1g,h**; **Suppl. Fig. 6a,b**, and **Fig. 6b,c** in the revised submission.

Minor issues:

Antibodies used for ChIP-seq are not listed in the methods section.

Response:

We apologize for not including this information into the first submission. All antibodies used in our study are listed in the “Reporting Summary” which were submitted to *Nature Communications* as a separate document.

Reviewers' Comments:

Reviewer #1:

Remarks to the Author:

We are pleased to see that the authors have addressed all comments very well. There are a couple of points where the authors are not sure whether to include or not the data. We would recommend:

- 1) to keep out the short-term YFP tracing, as suggested by the authors (rev fig 1)
- 2) to include the low magnification YFP tracing, showing the recombination efficiency (rev fig 3)

Reviewer #2:

Remarks to the Author:

The authors carefully addressed all my concerns. The work is addressing a timely topic and is well suited for publication in Nature Communications.

Reviewer #3:

Remarks to the Author:

This revised manuscript is improved, and the authors have addressed most of my previous concerns. Overall this study represents an important advance in the field providing a mechanistic basis for how the duration of transcriptional responses to extracellular signals are modulated and how LSD1 can epigenetically decommission enhancers after the removal of the signal. In addition to informing pancreas development this mechanism is likely to impact signal responsive transcription in many biological contexts.

Minor point that should be resolved: Showing that LSD1 and FOXA/GATA bind to the same enhancers is a nice addition, but it does not demonstrate that FOXA recruits LSD1 – the authors should temper this conclusion.

Reviewer #4:

Remarks to the Author:

The authors have been thoroughly attentive to the initial reviews, and their final version nicely describes the mechanism of LSD1 function in pancreatic endocrine development. Although there are still a few inconclusive areas, these are trivial overall compared to the strength of the data and their interpretation. I expect that this work will be of great interest to the pancreas development community, and I support its publication.

Reviewer #1 (Remarks to the Author):

We are pleased to see that the authors have addressed all comments very well. There are a couple of points where the authors are not sure whether to include or not the data. We would recommend:

- 1) to keep out the short-term YFP tracing, as suggested by the authors (rev fig 1)
- 2) to include the low magnification YFP tracing, showing the recombination efficiency (rev fig 3)

We included Rev Figure 3 as Supplementary Figure 9f.

Reviewer #3 (Remarks to the Author):

This revised manuscript is improved, and the authors have addressed most of my previous concerns. Overall this study represents an important advance in the field providing a mechanistic basis for how the duration of transcriptional responses to extracellular signals are modulated and how LSD1 can epigenetically decommission enhancers after the removal of the signal. In addition to informing pancreas development this mechanism is likely to impact signal responsive transcription in many biological contexts.

Minor point that should be resolved: Showing that LSD1 and FOXA/GATA bind to the same enhancers is a nice addition, but it does not demonstrate that FOXA recruits LSD1 – the authors should temper this conclusion.

Line 141: we changed to: “Distal LSD1 peaks at PP1 overlapped with binding sites for the early pancreatic TFs FOXA1, FOXA2, GATA4, GATA6, and HNF6, suggesting that these TFs reside in a complex with LSD1 (**Supplementary Fig. 5b,c**).”